## PROCEEDINGS A

applied mathematics, computational mathematics, statistics

random Fourier features, Metropolis algorithm, spatial interpolation, machine learning, wind field reconstruction, flow field estimation

**Author for correspondence:**
Emanuel Ström
e-mail: emastr@kth.se

# Wind field reconstruction with adaptive random Fourier features

Jonas Kiessling[1,2], Emanuel Ström[2] and
Raúl Tempone[3,4]

[1]H-Ai AB, Stockholm, Sweden
[2]KTH Royal Institute of Technology, Stockholm, Sweden
[3]RWTH Aachen University, Aachen, Germany
[4]KAUST, Thuwal, Saudi Arabia

ES, 0000-0001-7372-8535

We investigate the use of spatial interpolation methods for reconstructing the horizontal near-surface wind field given a sparse set of measurements. In particular, random Fourier features is compared with a set of benchmark methods including kriging and inverse distance weighting. Random Fourier features is a linear model $\beta(x) = \sum_{k=1}^{K} \beta_k \, e^{i\omega_k x}$ approximating the velocity field, with randomly sampled frequencies $\omega_k$ and amplitudes $\beta_k$ trained to minimize a loss function. We include a physically motivated divergence penalty $|\nabla \cdot \beta(x)|^2$, as well as a penalty on the Sobolev norm of $\beta$. We derive a bound on the generalization error and a sampling density that minimizes the bound. We then devise an adaptive Metropolis–Hastings algorithm for sampling the frequencies of the optimal distribution. In our experiments, our random Fourier features model outperforms the benchmark models.

## 1. Introduction

An integral part in wind farm planning and weather prediction is access to high-quality wind data [1,2]. However, meteorological stations are often heterogeneously distributed with many gaps, and interpolation techniques have to be employed in order to increase spatial resolution [3]. For example, a crucial step in building wind farms is *site prospecting*, in which national or state-level measurements are used to estimate the

expected aggregate yearly or monthly energy output [1]. A common approach is to approximate the probabilistic distribution of the wind speed over time with some parametric model, apply spatial interpolation to the parameters, calculate the energy output as a function of wind speed and then take the expected value. The work [4] lists approximately 200 papers written between 1940 and 2008 that focus on parametric models for time series of wind speed measurements. More modern approaches that directly interpolate wind speed also exist [3].

In this work, we focus on interpolation of the near-surface north-south and east-west horizontal wind components for short, 10-min time intervals. This effectively results in a reconstruction of the entire wind vector field, hence the name *wind field reconstruction*. Due to high spatial variability and dependence on both local and upstream terrain features, interpolation of wind data over hourly or minute-wise intervals is considered a particularly hard problem [3]. However, there are reasons why this can be useful, such as predicting the propagation of forest fires and pollutants, modelling the movement of flying animals and insects [5], or formulating initial conditions for atmospheric simulations and numerical weather prediction models [2,6]. In such applications, spatial interpolation is used to transfer measurements from meteorological stations to a fine grid or mesh.

There is relatively little research published on wind field reconstruction by the geological community [3]. Recent studies on machine learning interpolation methods like neural networks, support vector machines and random forest in other geological applications such as air temperatures [7], snow depth [8], mineral concentrations [9] and wind speed [4,10,11] have shown promising results. These types of models trade interpretability for power and flexibility and allow for efficient inclusion of covariates such as elevation, terrain slope, concavity and roughness [7]. This raises the question of whether machine learning-based interpolation works for wind field reconstruction as well. In a 2018 study by Reinhardt & Samimi [3], the established method of regression kriging was shown to reliably outperform some neural networks and support vector machines in horizontal wind field reconstruction. The authors note that the performance of machine learning methods was considerably worse for near surface winds, and argue that the models are unable to capture the complex behaviour and high variability that arises from interactions between wind and terrain.

Since the 2018 study by Reinhardt & Samimi, there have been significant advancements in the related field of fluid flow reconstruction. In particular, approaches based on generalizations of principal component decomposition have been trained to interpolate sparse measurements directly to high resolution grids with incredibly high accuracy [12–14]. The difference between these methods and traditional interpolation is that the training data consist of measurement from multiple times. However, the resolution of these methods can only be as high as the training data. High-resolution data are sometimes not available, or might take a considerable time and effort to simulate. In such cases, simpler interpolation models can arguably be a valid alternative. The definition of an interpolation model varies depending on the context. The definition used in our work draws from [9], and is synonymous with regression.

In this paper, we compare a novel machine learning method known as random Fourier features with a selection of popular and successful interpolation techniques. The model fits a Fourier series $\beta(x) = \sum_k \beta_k e^{i\omega_k x}$ to the data. Instead of traditional greedy optimization methods such as stochastic gradient descent, our random Fourier features model explores the frequency domain using an adaptive Metropolis algorithm inspired by [15]. In each step, the Fourier coefficients $\beta_k$ are optimized with respect to a loss function. The work [16] proposes a technique for interpolation of incompressible flow by incorporating zero divergence as a constraint in the optimization. We propose an alternative approach where the incompressibility takes the form of a regularization term.

The method presented in this report only considers data from a fixed time point and does not model the wind flow through time. The method can be extended to model the flow for instance with the principal component technique explained in [13,17]. The article is structured as follows. In §2, a mathematical formulation of the wind field reconstruction problem is formulated. The data are also presented, along with error estimates used to evaluate the models. Section 3

contains a short introduction to the interpolation models. The results are presented in §4 and finally, discussed in §5.

## 2. Problem formulation

Let $\Omega \subset \mathbb{R}^2$ denote a geographical region. For the purpose of this report, $\Omega$ is the set of points contained within the Swedish borders. We define the horizontal wind field $\boldsymbol{u} : \Omega \times [0, \infty) \to \mathbb{R}^2$ that maps every point in space $\boldsymbol{x} = (x, y) \in \Omega$ and time $t > 0$ to a velocity vector $\boldsymbol{u}(\boldsymbol{x}, t) = (u(\boldsymbol{x}, t), v(\boldsymbol{x}, t)) \in \mathbb{R}^2$. In mesoscale atmospheric models, a set of physical assumptions involving quantities like the wind field, temperature and pressure are used to pose a set of differential equations that describe the time evolution of the system. Given an initial state at the time $t = 0$ and a set of boundary conditions, it is possible to simulate and forecast the wind field $\boldsymbol{u}$ using these equations. The spatial interpolation methods considered in this work do not take previous or future times into account. Thus, only a subset of the physical assumptions about the system can effectively be enforced. Some of the presented models will incorporate the assumption of divergence-free flow, which is well established in mesoscale atmospheric models [6]. It can be written as a differential equation $\nabla \cdot \boldsymbol{u} = 0$, where $\nabla \cdot \boldsymbol{u} = \partial u / \partial x + \partial v / \partial y$ is the divergence of the wind field $\boldsymbol{u}$. The above notation as well as the notation presented in §2a,b will be used extensively throughout our work.

### (a) Data

The data were obtained from the Swedish Meterological and Hydrological Institute. It contains a set of wind observations during the entirety of 2018, from $N = 171$ weather stations scattered across Sweden as shown in figure 1. Each measurement is collected 10 m above ground. The positions of the stations are given by the elevation—the height above sea level—as well as the latitude and longitude. The measurements are 10-min averages, collected once every hour. They consist of two components: the wind speed, which is the magnitude of the horizontal component of the wind vector, and the wind direction, which is the angle from which the wind comes, measured clockwise relative to north ($90°$ is east, $180°$ is south and so on). Both measurements are rounded to varying degrees of precision depending on the station [18]. The stations are not active at all times. In fact, there are occasional hours with as few as one station reporting measurements. The wind measurements are highly autocorrelated over time, as demonstrated by the autocorrelation plots of the south-north and east-west wind components in figures 2 and 3. The data were processed before training. First, the latitude–longitude pairs were transformed to Cartesian coordinates $\boldsymbol{x} = (x, y)$ where $x$ is the eastward-measured distance and $y$ is the northward measured distance, as shown in figure 1. This was done using the SWEREF 99 TM[1] map projection. Second, the wind measurements were transformed from polar to Cartesian coordinates $\boldsymbol{u} = (u, v)$ forming a horizontal wind vector, where $u$ corresponds to the velocity component along the $x$-axis and $v$ corresponds to the velocity component along the $y$-axis. Lastly, all wind measurements from the month of September were removed. The measurements exhibited a root mean square wind speed of $5.5\,\mathrm{m\,s^{-1}}$ as opposed to $4.5\,\mathrm{m\,s^{-1}}$ for the remaining months, meaning that the error metrics were disproportionally affected by September. Furthermore, the wind field exhibited low spatial variability during this period. These highly uniform, low variance fields are not the main interest of the study. The removal of the September data did not affect the method comparison significantly.

### (b) Spatial interpolation models

For the purpose of this report, the definition of interpolation models is restricted to approximations of functions $\boldsymbol{u} : \Omega \to \mathbb{R}^2$. The range is two-dimensional since we are modelling the

---

[1]More information about the SWEREF 99 TM map projection can be found here: www.lantmateriet.se/en/maps-and-geographical-information/gps-geodesi-och-swepos/Referenssystem/Tvadimensionella-system/SWEREF-99-projektioner.

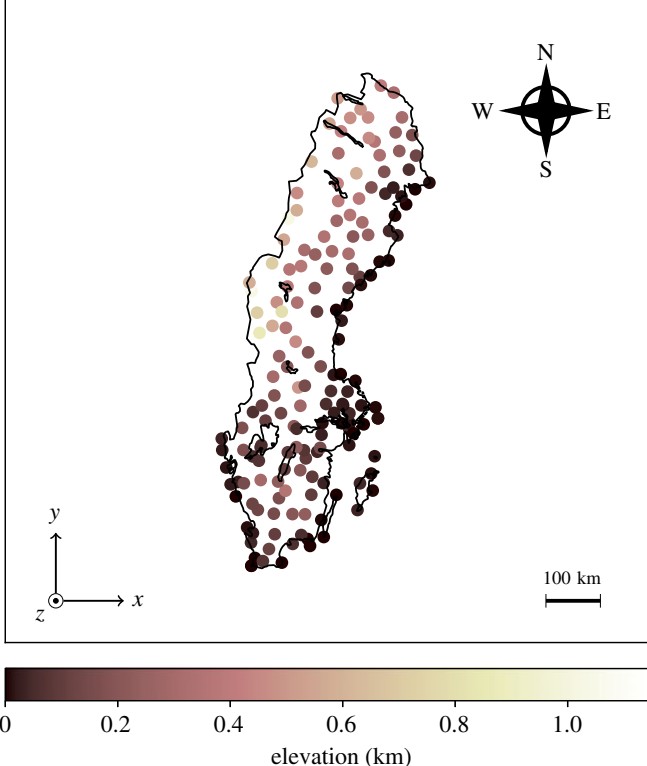

**Figure 1.** The `SWEREF 99 TM` orthographic projection of Sweden. The weather stations are marked with circles, coloured according to their elevation $z$. The coordinate system is drawn in the bottom left. The centre of the map is at 62°40′6″ N, 17°55′19″ E. (Online version in colour.)

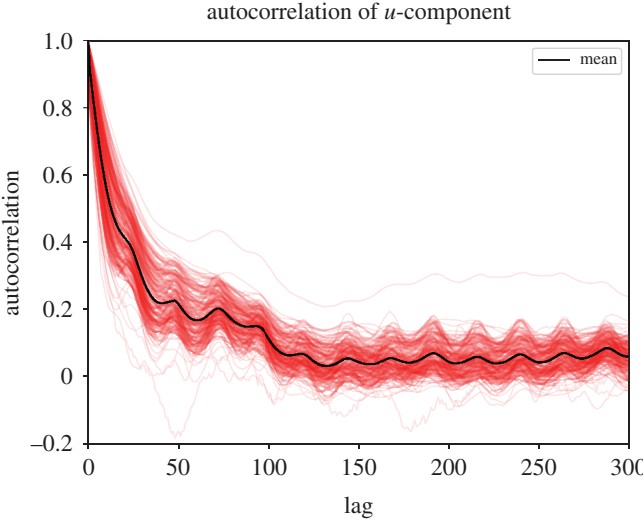

**Figure 2.** Autocorrelation of the east-west component $u$ of the wind $\boldsymbol{u}$ for the available weather stations, up to a lag of 300 hours. Each red line represents the autocorrelation of one weather station. (Online version in colour.)

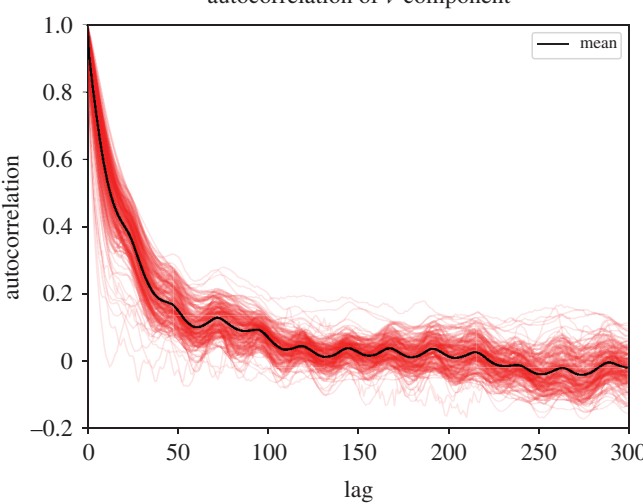

**Figure 3.** Autocorrelation of the north-south component $v$ of the wind $\boldsymbol{u}$ for the available weather stations, up to a lag of 300 hours. Each red line represents the autocorrelation of one weather station. (Online version in colour.)

horizontal wind vector field, and the region $\Omega$ is Sweden. In general, interpolation models are also used for other properties like temperature, pressure and population density, etc. Nevertheless, we define a *spatial interpolation model* as a map $f$ from a set of velocity measurements $\mathcal{D} = \{(\boldsymbol{x}_n, \boldsymbol{u}_n) : \boldsymbol{x}_n \in \Omega, \ n = 1, 2, \dots, N\}$ to a vector field $f_{\mathcal{D}} : \Omega \to \mathbb{R}^2$. The process of evaluating a model $f$ on $\mathcal{D}$ is called *training*. Usually, the training is done by minimizing a loss function. The above notation will be used throughout our report. Note that there is an important distinction between $f$ and $f_{\mathcal{D}}$. The function $f$ represents a model, which is trained on data $\mathcal{D}$ and produces a vector field $f_{\mathcal{D}}$ that approximates $\boldsymbol{u}$. Any time a symbol is indexed by the letter $\mathcal{D}$ or variations of it such as $\mathcal{D}_t$, it symbolizes an interpolation model trained on that particular dataset. A parametrized set of interpolation models $\{f^\theta : \theta \in \Theta\}$ of interpolation models $f^\theta$ is called an *interpolation family*, and the parameters $\theta$ are called *hyper parameters*. For example, the $K$-nearest neighbours models can be viewed as an interpolation family with hyper parameter $K$. The process of finding the best model $f^\theta$ in an interpolation family with respect to some quality of fit is called *hyper optimization*.

## (c) Quality of fit

Given an interpolation model $f$ and observations $\{\mathcal{D}_t\}_{t \in T}$ at different times $t$ in a time span $T$, the quality of fit $\mathcal{Q}(f)$ is defined as the expected square loss with respect to the distribution $\rho(\mathrm{d}\boldsymbol{x}, \mathrm{d}t)$ of the data in space and time $\Omega \times T$:

$$\mathcal{Q}(f) = \mathbb{E}[||f_{\mathcal{D}_t}(\boldsymbol{x}) - \boldsymbol{u}(\boldsymbol{x}, t)||^2] = \int_T \int_\Omega ||f_{\mathcal{D}_t}(\boldsymbol{x}) - \boldsymbol{u}(\boldsymbol{x}, t)||^2 \rho(\mathrm{d}\boldsymbol{x}, \mathrm{d}t). \qquad (2.1)$$

We only have access to a limited set of data, both in space and time. Hence, calculating the exact value of $\mathcal{Q}$ is not possible. Furthermore, using all available data to compute $\mathcal{Q}$ is too computationally expensive. Instead, we use a Monte Carlo sampling average in time and a cross-validation scheme in space. Let $t_1, t_2, \dots, t_{N_T}$ be all times in $T$ for which measurements exist, that is every full hour for the entirety of 2018, excluding the entirety of September. Next, we create a set $\mathcal{K}$ of indices by sampling randomly from the integers 1 to $N_T$ with replacement. Then, define a set $\{t_k\}_{k \in \mathcal{K}}$ of time samples and let $\mathcal{D}_k = \{(\boldsymbol{x}_n, \boldsymbol{u}_{kn}) : n = 1, 2, \dots, N_k\}$ be the set of observations made at the time $t_k$. As stated in §2a, the weather stations are not always active, so the number of measurements $N_k$ at the time $t_k$ varies. Define a random partition of the weather stations into $M$ disjoint sets of equal size $\mathcal{D}_k^m$, $m = 1, 2, \dots, M$, and denote $\mathcal{D}_k^{-m} = \mathcal{D}_k \backslash \mathcal{D}_k^m$. The

fixed time expectations $\mathcal{Q}_t(f) = \mathbb{E}[||f_{\mathcal{D}_t} - u(x,t)||^2 \mid t]$ of the quality of fit at the sampled times $t_k$ are approximated using cross-validation:

$$\mathcal{Q}_{t_k}(f) \approx \widetilde{\mathcal{Q}}_{t_k}(f) := \frac{1}{|\mathcal{D}_k|} \sum_{m=1}^{M} \sum_{(x,u) \in \mathcal{D}_k^m} ||f_{\mathcal{D}_k^{-m}}(x) - u||^2. \tag{2.2}$$

The above formula can be summarized by the following procedure. For every split $m$, train the model $f$ on the data $\mathcal{D}_k^{-m}$ and calculate the square errors on the remaining points $\mathcal{D}_k^m$. Then, take the mean of all squared errors. We found that fivefold cross-validation ($M = 5$) struck a good balance between computation time and accuracy. Averaging $\widetilde{\mathcal{Q}}_{t_k}(f)$ over the samples in $\mathcal{K}$ yields a Monte Carlo sample estimate of $\mathcal{Q}(f)$:

$$\mathcal{Q}(f) \approx \widetilde{\mathcal{Q}}(f) := \frac{1}{|\mathcal{K}|} \sum_{k \in \mathcal{K}} \widetilde{\mathcal{Q}}_{t_k}(f), \tag{2.3}$$

also called the *variance unexplained*. An alternative measure $\mathcal{E}$ is obtained from normalizing $\mathcal{Q}(f)$ by the expected square wind speed $\mathbb{E}[||u(x,t)||^2]$, denoted $\mathcal{Q}(0)$:

$$\mathcal{E}(f) = \frac{\mathcal{Q}(f)}{\mathcal{Q}(0)} \approx \frac{\widetilde{\mathcal{Q}}(f)}{\widetilde{\mathcal{Q}}(0)} =: \widetilde{\mathcal{E}}(f). \tag{2.4}$$

This measurement is referred to as the *fraction of variance unexplained*. If the wind field $u$ has zero mean, the number $1 - \mathcal{E}(f)$ is referred to as $\mathcal{R}^2$, or the *coefficient of determination*. We estimate the variance of $\widetilde{\mathcal{Q}}$ as

$$\text{Var}(\widetilde{\mathcal{Q}}) = \frac{\text{Var}(\mathcal{Q}_t)}{|\mathcal{K}|} \quad \text{and} \quad \text{Var}(\mathcal{Q}_t) \approx \frac{1}{|\mathcal{K}|} \sum_{k \in \mathcal{K}} (\widetilde{\mathcal{Q}}_{t_k} - \widetilde{\mathcal{Q}})^2. \tag{2.5}$$

In §4, we show that the variance unexplained approximately follows a normal distribution. Consequently, two standard deviations of $\widetilde{\mathcal{Q}}$ constitute an approximate 95% confidence interval for $\mathcal{Q}$. Note that the confidence bound for $\mathcal{Q}$ is not sufficient when comparing models, since the errors can be highly correlated. Given two models $f, g$ we instead define the difference $\Delta\mathcal{Q}(f,g) := \mathcal{Q}(f) - \mathcal{Q}(g)$ and estimate it as $\Delta\widetilde{\mathcal{Q}}(f,g) := \widetilde{\mathcal{Q}}(f) - \widetilde{\mathcal{Q}}(g)$. The variance of $\Delta\widetilde{\mathcal{Q}}(f,g)$ is estimated similarly to (2.5):

$$\text{Var}(\Delta\widetilde{\mathcal{Q}}(f,g)) = \frac{\text{Var}(\Delta\mathcal{Q}_t(f,g))}{|\mathcal{K}|}, \quad \text{Var}(\Delta\mathcal{Q}_t(f,g)) \approx \frac{1}{|\mathcal{K}|} \sum_{k \in \mathcal{K}} (\Delta\widetilde{\mathcal{Q}}_{t_k}(f,g) - \Delta\widetilde{\mathcal{Q}}(f,g))^2. \tag{2.6}$$

The sample size $|\mathcal{K}|$ was adjusted to obtain 95% confidence intervals within 1% of the mean square wind speed on the 2018 wind dataset, for both the variance unexplained $\mathcal{Q}(f)$ and difference in variance unexplained $\Delta\mathcal{Q}(f,g)$. Due to high autocorrelation in the wind demonstrated in figure 2 as well as the presence of seasonality and trends, these confidence intervals do not generalize to longer time intervals than 2018. The quality of fit was used for both hyper parameter optimization and validation of the models, but with different data samples to avoid overfitting.

# 3. Models

## (a) Fourier models

In this section, we introduce two Fourier series-based models. At the end of the section, we arrive at the random Fourier features model, which is the main focus of our report. In §3b, we present some well established and high-performing interpolation models. We use these as benchmark models when assessing the interpolation quality of the random Fourier features.

## (i) Fourier series

The Fourier series-based model takes the form

$$\beta(x) = \Re \left\{ \sum_{k=1}^{K} \widehat{\beta}_k \, e^{i\omega_k \cdot x} \right\}, \quad \omega_k \in \mathbb{R}^2, \ \widehat{\beta}_k \in \mathbb{C}^2, \ k = 1, 2, \ldots, K, \tag{3.1}$$

where $K$ is the number of terms, $\omega_k \cdot x$ denotes the scalar product between $\omega_k$ and $x$ and $\Re\{z\}$ denotes the real part of the complex-valued vector $z$. The Fourier series generalizes to arbitrary dimensions of the input $x$, but in this report we chose $x = (x, y)$ to be just the horizontal coordinates. With slight abuse of notation, we let $\widehat{\beta}_k$ and $\omega_k$ denote two-dimensional vectors in $\mathbb{C}^2$ and $\mathbb{R}^2$ respectively. Furthermore, $\omega = (\omega_1, \omega_2, \ldots, \omega_K)$ and $\widehat{\beta} = (\widehat{\beta}_1, \widehat{\beta}_2, \ldots, \widehat{\beta}_K)$. In the Fourier series-based model, $\omega$ is held fixed, and the parameters $\widehat{\beta}$ are estimated by optimizing with respect to the expectation of a loss function $\ell : \mathbb{R}^2 \times \mathbb{R}^2 \to \mathbb{R}^+$:

$$\min_{\widehat{\beta}} \quad \mathbb{E}_{\tilde{\rho}}[\ell(\beta(x), u)], \tag{3.2}$$

where the expectation is taken over the joint density $\tilde{\rho}$ of the data $x, u$. For our application, $\tilde{\rho}$ is unknown. Therefore, the expected loss is replaced by a Monte Carlo sample estimate of the loss, called the *empirical loss*. Given a dataset $\mathcal{D} = \{(x_n, u_n)\}_{n=1}^{N}$, the empirical loss is expressed as

$$\frac{1}{N} \sum_{n=1}^{N} \ell(\beta(x_n), u_n) \approx \mathbb{E}_{\tilde{\rho}}[\ell(\beta(x), u)]. \tag{3.3}$$

For this work, we chose the following loss function:

$$\ell(\beta(x), u) = ||\beta(x) - u||^2 + \lambda ||\beta||^2_{S(2,2)} + \eta ||\nabla \cdot \beta||^2_{L^2}. \tag{3.4}$$

The expression $||\beta||^2_{S(2,2)}$ denotes the second-order squared Sobolev-norm of $\beta$ [19]:

$$||\beta||^2_{S(2,2)} = ||\beta||^2 + r(||\partial_x \beta||^2 + ||\partial_y \beta||^2) + r^2(||\partial_{xx}\beta||^2 + 2||\partial_x \partial_y \beta||^2 + ||\partial_{yy}\beta||^2), \tag{3.5}$$

where $r > 0$ is a hyper parameter. By orthogonality of the Fourier features, the Sobolev norm can be simplified to

$$||\beta||^2_{S(2,2)} = \sum_{k=1}^{K} (r^2 ||\omega_k||^4 + r ||\omega_k||^2 + 1) ||\widehat{\beta}_k||^2. \tag{3.6}$$

The hyper parameter $r$ was added to allow for more flexibility in the penalty, and is equivalent to rescaling the input variable before applying the Sobolev norm. The expression $||\nabla \cdot \beta||^2_{L^2}$ is the squared $L^2$-norm of the divergence of $\beta$, which can be rewritten as

$$||\nabla \cdot \beta||^2_{L^2} = \sum_{k=1}^{K} |\omega_k \cdot \widehat{\beta}_k|^2. \tag{3.7}$$

The intended effect of the Sobolev norm is to dampen high frequencies, and the divergence penalty is supposed to simulate incompressible flow. The empirical loss using the loss function as defined in (3.4) is

$$\frac{1}{N} \sum_{n=1}^{N} ||\beta(x_n) - u_n||^2 + \lambda ||\beta||^2_{S(2,2)} + \eta ||\nabla \cdot \beta||^2_{L^2}. \tag{3.8}$$

Here, $\lambda$ and $\eta$ are hyper parameters. In order to determine a suitable choice for $\omega$, assume the standard regression setting $u = f(x) + \epsilon$, where $\epsilon$ is zero mean and independent of $x$. Since the spatial region $\Omega$ is bounded, $f$ can be extended periodically over $\mathbb{R}^2$. That is, the relation $f(x + m\tau_x, y + n\tau_y) = f(x, y)$ is imposed on $f$ for all $x = (x, y) \in \mathbb{R}^2$, whole numbers $m, n$ and some two-dimensional period $\tau = (\tau_x, \tau_y)$. This means that $f$ can be written as a Fourier series. Thus, $f$ can in theory be approximated arbitrarily well by $\beta$ as the number of terms $K$ tends to infinity. However, since there is only a limited amount of data and $\beta$ contains a limited number of terms, the choice

of $\omega$ determines how well $\beta$ can approximate $f$. In the Fourier series-based model, we settle on choosing $\omega$ as a square grid with side length $2M+1$, centred at the origin in the frequency domain:

$$\omega = \left\{ \left( \pi \frac{m}{\tau_x}, \pi \frac{n}{\tau_y} \right) : -M \le m \le M, \quad -M \le n \le M \right\}. \tag{3.9}$$

As such, the Fourier series-based model is a spatial interpolation family with the hyper parameters $\lambda$, $\eta$, $M$ and $\tau$. We found $M = 10$ struck a balance between computational cost and accuracy. The remaining hyper parameters $\lambda$, $\eta$, $\tau$ and $r$ were chosen as explained in the next section. Minimizing the expression in (3.8) amounts to solving a system of linear equations with respect to the elements $\beta$.

## (ii) Random Fourier features

Instead of optimizing with respect to $\widehat{\boldsymbol{\beta}} = (\widehat{\beta}_1, \widehat{\beta}_2, \dots, \widehat{\beta}_K)$, random Fourier features aims at solving the harder problem of also optimizing with respect to the Fourier frequencies $\omega = (\omega_1, \omega_2, \dots, \omega_K)$:

$$\min_{\boldsymbol{\beta}, \omega} \quad \mathbb{E}[\ell(\beta(x), u)]. \tag{3.10}$$

The random Fourier features is an example of a neural network with one hidden layer and trigonometric activation function [15]. Here, $\omega$ are the weights connecting the inputs $x$ to the hidden layer, and $\widehat{\boldsymbol{\beta}}$ are the weights connecting the nodes in the hidden layer to the output layer. What distinguishes random Fourier features is the training algorithm. The training of neural networks is traditionally done using some greedy method such as stochastic gradient descent, but in random Fourier features, the frequencies $\omega$ are assumed to be randomly sampled according to some distribution $\rho$. The task of optimizing frequencies is thus changed to optimizing $\rho$. Note that the distribution can be deterministic, so no approximation error is introduced by this. Next, the optimization of $\widehat{\boldsymbol{\beta}}$ is moved inside the frequency expectation

$$\min_{\widehat{\boldsymbol{\beta}}, \rho} \mathbb{E}[\ell(\beta(x), u)] \le \min_{\rho} \mathbb{E} \left[ \min_{\widehat{\boldsymbol{\beta}}} \mathbb{E}[\ell(\beta(x), u) \mid \omega] \right]. \tag{3.11}$$

This reduces the inner minimization problem to linear regression with Fourier features. Using the quadratic loss function defined in (3.4), the coefficients $\widehat{\boldsymbol{\beta}}$ can be found using a standard matrix inversion.

The key approximation of random Fourier features is to assume that the elements of $\omega$ are independent and identically distributed according to a density $\rho$. That is, $\boldsymbol{\rho}(\omega) = \prod_{k=1}^{K} \rho(\omega_k)$. The optimal $\rho$ is approximated by finding an analytical minimizer to an upper bound of (3.10). We assume the same standard regression setting $u = f(x) + \epsilon$ as presented in the previous section, where $\epsilon \in \mathbb{R}^2$ is zero mean and independent of $x$ and $f$ a periodic function. Referring to proposition A.1 found in the appendix, we can then put an upper bound

$$\min_{\rho} \mathbb{E} \left[ \min_{\widehat{\boldsymbol{\beta}}} \mathbb{E}[\ell(\beta(x), u) \mid \omega] \right]$$

$$\le \frac{1 + \lambda \overline{C}_1 + \eta \overline{C}_2}{(2\pi)^2 K} \sqrt{\mathbb{E}\left[ \frac{||\hat{f}(\omega)||^4}{\rho(\omega)^4} \right]} - \frac{1}{K} \mathbb{E}[||f(x)||^2] + \mathbb{E}[||\epsilon||^2], \tag{3.12}$$

on the expected loss, where $\overline{C}_1$ and $\overline{C}_2$ are positive constants determined by the Sobolev and divergence loss functions, respectively, and $\omega$ is a random variable distributed according to $\rho$, representing an arbitrary element of $\omega$. The proof assumes that the distribution $\rho$ is discrete and has bounded moments up to a degree determined by the order of the derivatives used in the regularization. Furthermore, we show that this upper bound is minimized by choosing $\rho(\omega) \propto ||\hat{f}(\omega)||$, where $\hat{f}(\omega)$ are the Fourier coefficients for the Fourier series expansion of $f$.

We make two important comments about the limitations of this method. First, note that we are only minimizing the bound of the iterated expectation in (3.12). Second, the iterated expectation

is also only a bound for the true loss function (3.11). Despite these limitations, random Fourier features has in some examples been shown to outperform traditional methods of optimizing (3.10) like stochastic gradient descent, possibly due to its ability to efficiently learn high-frequency details early on in the training [15].

Since the target distribution $\rho(\omega) \propto ||\hat{f}(\omega)||$ is not known *a priori*, this distribution has to be approximated somehow. The authors of [15] present an adaptive Metropolis algorithm for sampling from $\rho$ in a related setting. The main differences are that $f$ is $L^2$ integrable on the entirety of $\mathbb{R}^2$ and has a one-dimensional range. Drawing from the work in [15], we devise an adaptive Metropolis algorithm for sampling the frequencies $\boldsymbol{\omega}$ from $||\hat{f}||$. This algorithm is described in detail below (algorithm 1).

---

**Algorithm 1** . Discrete random Fourier features with Metropolis sampling

---

**input** : Rescaled data $\{x_n, u_n\}_{n=1}^N \subset [0,1]^2 \times \mathbb{R}^2$
**output:** Random features $x \mapsto \sum_{k=1}^K \widehat{\beta}_k \, \mathrm{e}^{\mathrm{i}\omega_k \cdot x}$
$K \leftarrow$ Choose the number of frequencies;
$B \leftarrow$ Choose the number of steps;
$\sigma \leftarrow$ Choose a variance for the proposal kernel;
$\gamma \leftarrow$ Choose the exponent for the transition probability;
$\lambda \leftarrow$ Choose the Sobolev regularization parameter;
$\eta \leftarrow$ Choose the divergence regularization parameter;
$\boldsymbol{\omega} \leftarrow$ the zero vector in $\mathbb{R}^{2K}$;
$\widehat{\boldsymbol{\beta}} \leftarrow$ minimizer of the empirical (3.8) loss given $\boldsymbol{\omega}$;
**for** $b \leftarrow 1, \dots, B$ **do**
 $\delta_{\mathcal{N}} \leftarrow$ standard normal random vector in $\mathbb{R}^{2K}$;
 $\delta \leftarrow$ round the elements of $\sigma\delta_{\mathcal{N}}$ to the nearest integer;
 $\boldsymbol{\omega}' \leftarrow \boldsymbol{\omega} + \delta$;
 $\widehat{\boldsymbol{\beta}'} \leftarrow$ minimizer of the empirical loss (3.8) given $\boldsymbol{\omega}'$;
 **for** $k \leftarrow 1, \dots, K$ **do**
 $\alpha \leftarrow$ sample from uniform distribution on $[0,1]$;
 **if** $||\widehat{\beta}'_k||^\gamma / ||\widehat{\beta}_k||^\gamma > \alpha$ **then**
 $\omega_k \leftarrow \omega'_k$;
 $\widehat{\beta}_k \leftarrow \widehat{\beta}'_k$;
 **end**
 **end**
**end**
$\widehat{\boldsymbol{\beta}} \leftarrow$ minimizer of the empirical loss (3.8) given $\boldsymbol{\omega}$;
$x \mapsto \sum_{k=1}^K \widehat{\beta}_k \, \mathrm{e}^{\mathrm{i}\omega_k \cdot x}$

---

The hyper parameters of the random Fourier features consist of the Fourier frequencies $K$, the periodicity $\tau$, the regularization parameters $\eta, r$ and $\lambda$—these are also found in the Fourier series model—as well three new parameters that appear in the Metropolis sampling algorithm: the number of steps $B$, the standard deviation $\sigma$ of the random walk and a parameter $\gamma$ that adjusts the acceptance probability of each step. Similarly to the work [16], the period $\tau$ was chosen equal in both cardinal directions, about two times the north-south length of the region of interest: $\tau_x = \tau_y = 4000$ km. The regularization parameter $r$ was set to $1/\tau$ in order to prevent the second derivative from dominating the Sobolev regularization term. The number of frequencies was fixed to 400 and the number of steps $B$ was fixed to 500. Given a $d$-dimensional feature space and a Gaussian target distribution, [15] show that for a fixed computational cost, the upper bound (3.12) as $K, N \to \infty$ is approximately minimized by $\gamma = 3d - 2$. Furthermore, the classical result from

[20] shows that the optimal variance of the proposal kernel for a general random walk Metropolis–Hastings algorithm is $\sigma^2 \approx 2.4^2/d$.

The regularization parameters $\lambda$ and $\eta$ were obtained by running a grid search to minimize the variance unexplained in (2.4), while keeping $\tau$, $r$, $\gamma$ and $\sigma$ fixed as specified above. Since the values for $\gamma$ and $\sigma$ are only optimal in the limit, a second grid search was done in a neighbourhood of the limit values for $\gamma$ and $\sigma$, while keeping $\tau$, $r$, $\lambda$ and $\eta$ fixed. The number of steps was adjusted to ensure convergence of the Metropolis algorithm to a stationary distribution, while also keeping the computation time low. Note that the random Fourier features algorithm is not guaranteed to find the optimal frequencies. There is ongoing research in this area, and an iterative greedy method is explored in [17].

## (b) Benchmarking models

The following interpolation models were used for benchmarking: nearest neighbours, inverse distance weighting (IDW), kriging, random forest, neural networks and a Fourier series-based model introduced in the previous section. This section will serve as a short introduction to each of the methods as well as motivation as to why they are relevant. We use the same notation $\mathcal{D} = \{(x_n, u_n) \in \Omega \times \mathbb{R}^2 : n = 1, 2, \ldots, N\}$ for the measurements as in §2c.

### (i) IDW

IDW methods $\{f^p : p \geq 0\}$ is an interpolation family of methods that are evaluated at a point $x$ by taking a component-wise weighted average of the horizontal wind vector data in $\mathcal{D}$. Specifically, the weights $\alpha(x_n, x)$ for a model $f^p$ from the IDW family are proportional to $1/d(x_n, x)^p$ where $d : \Omega \times \Omega \to [0, \infty)$ is a distance on $\Omega$. That is,

$$f_{\mathcal{D}}^p(x) = \frac{\sum_{n=1}^N \alpha(x_n, x) u_n}{\sum_{n=1}^N \alpha(x_n, x)}, \quad \text{where } \alpha(x_n, x) = \frac{1}{d(x_n, x)^p}. \tag{3.13}$$

The singularities at $x = x_n$, $n = 1, 2, \ldots, N$ are removed by setting $f_{\mathcal{D}}^p(x_n) = u_n$. The hyper parameter $p$ adjusts the amount of influence each data point has over its immediate surroundings. Letting $p = 0$ will result in all points weighing equally everywhere, i.e. taking the arithmetic mean of the data. Letting $p \to \infty$ will result in the nearest neighbour method. Usually, $p$ is chosen somewhere in between. A common shortcoming of IDW is that the interpolated values are bounded by the maximum and minimum values of the data and therefore IDW fails to predict unobserved extreme points. The main benefits are interpretability and relatively short training time. In this report, two values of $p$ were tested, namely $p = 2$ and $p = \infty$ (i.e. nearest neighbours). Furthermore, we used the horizontal distance between points, ignoring elevation.

### (ii) Kriging

Kriging is a statistical approach to spatial interpolation [21]. The true velocity $u$ is assumed to satisfy the equality

$$u(x) = \mu(x) + \epsilon_x, \tag{3.14}$$

where $\mu : \Omega \to \mathbb{R}^2$ is a deterministic function and $\epsilon_x$ is a constant-mean, stochastic process over $x$. Although it is possible to model correlations between the vector components of $u$, we treat the two as independent, and run separate kriging pipelines for each component. For the remainder of this subsection, we therefore assume that the residual $\epsilon$ is in $\mathbb{R}$. The key idea in kriging is to endow the residuals with a specific translation-invariant structure. Namely, it is assumed that the variance of the difference $\epsilon_x - \epsilon_{x'}$ between two arbitrary residuals measured at points $x$ and $x'$ depends only on the distance $||x - x'||$ through a function called the *variogram*:

$$\mathbb{E}[(\epsilon_x - \epsilon_{x'})^2] = \gamma(||x - x'||). \tag{3.15}$$

The process of training a kriging model consists of two steps. First, a deterministic model is used to estimate the mean $\mu$ from the data, resulting in some trained model $\mu_{\mathcal{D}}$. For each $(x_n, u_n) \in$

$\mathcal{D}$, the residuals $\epsilon_{x_n}$ are estimated as $\epsilon_{x_n} \approx u_n - \mu_{\mathcal{D}}(x_n)$. The second training step then consists of fitting a variogram $\gamma_{\mathcal{D}}$ to the residuals $\epsilon_{x_n}$. Evaluating the model on a specific point $x$ can then be formulated as solving a minimization problem involving the variogram. Namely, kriging seeks a linear combination $\sum_n \omega_n \epsilon_{x_n}$ such that the weights $\omega$ minimize the expected square error $\mathbb{E}[(\epsilon_x - \sum_n \omega_n \epsilon_{x_n})^2]$. In order to get an unbiased estimate, the minimization is often subjected to a constraint $\sum_n \omega_n = 1$. Solving the constrained minimization problem results in a linear system of $N + 1$ equations

$$\sum_{m=1}^{N} \gamma_{\mathcal{D}}(||x_n - x_m||)\omega_m + \lambda = \gamma_{\mathcal{D}}(||x_n - x||), \quad n = 1, 2, \ldots, N \tag{3.16}$$

and

$$\sum_{m=1}^{N} \omega_m = 1, \tag{3.17}$$

where $\lambda$ is the Lagrange multiplier, an artificial variable that is brought in to enforce the constraint (3.17). Solving this system of equation produces a $N \times 1$ vector of weights $\omega_n(x)$, $n = 1, 2, \ldots, N$, which can be used to provide the final estimation $f_{\mathcal{D}}$ of $u$:

$$f_{\mathcal{D}}(x) = \mu_{\mathcal{D}}(x) + \sum_{n=1}^{N} \epsilon_{x_n} \omega_n(x). \tag{3.18}$$

A key strength of kriging is that the statistical model allows for point-wise error estimation. Furthermore, kriging can be combined with virtually any other unbiased deterministic interpolation model by interpreting it as the mean $\mu(x)$. However, when the statistical assumptions do not hold, kriging can perform poorly. Machine learning methods such as random forests have been successful in beating kriging for various spatial interpolation tasks, see for example [7,9]. The authors in [9] argue that even though kriging might be redundant in terms of accuracy, it remains a valuable tool for understanding data, exactly because of its statistical properties and interpretability. We used a version of kriging called *Universal kriging*, which differs from kriging in that a linear regression approximation of the mean $\mu(x)$ and the residuals $\epsilon_x$ are fitted to the data simultaneously, resulting in a joint system of equations for all the weights. The Python package `pykrige` was used to implement Universal kriging with a linear variogram and a piece-wise linear mean.

### (iii) Random forest

Random forests are constructed by averaging an ensemble of regression trees. A regression tree is a function that recursively splits the domain of interest into smaller domains, based on a user-specified criteria. Each domain is then assigned a constant value by minimizing some objective function on the training data [22]. In a random forest, each tree is trained on a random sample of the data $\mathcal{D}$ and each split in the tree is chosen by randomly selecting one feature out of the input features and picking a split that minimizes the mean square error [23]. Random forests have been used successfully in spatial interpolation problems, for example to predict temperatures on and around Kilimanjaro, Tanzania [7] as well as mineral concentrations [9]. The main drawback of random forests is lack of interpretability.

In this report, we used a random forest with 200 trees with mean square loss for splitting, and unlimited tree depth. The forest was implemented in Python using the package `scikit-learn`. Furthermore, the random forest was trained on a polynomial feature map $\phi$ of the horizontal coordinates $x = (x, y)$ and the elevation $z$. That is, the trained model $f_{\mathcal{D}}$ is a composition $h \circ \phi$ of the random forest $h_{\mathcal{D}}$ and the feature map $\phi$, which is fixed. The feature map increases expressivity of the random forest, which can improve performance if the data are not too noisy or sparse. There are many possible ways to construct feature maps. We found that a combination of polynomial features $x^{p_1} y^{p_2} z^{p_3}$ with a total order $p_1 + p_2 + p_3$ of $\leq 3$ struck a good balance between expressivity and generalizability.

### (iv) Feedforward neural network

Feedforward neural networks have been used extensively in different areas of applied mathematics. To read more about neural networks, see for example [24]. In this report, only a specific family of feedforward neural networks was considered. Namely, the networks are characterized by three fully connected hidden layers, a constant number of $n$ nodes in each layer and the ReLU activation function. The input layer consists of the three spatial coordinates $x, y$ and $z$. The weights were optimized using the Adam algorithm [25], with respect to the $L_2$-regularized loss. The network was implemented in `TensorFlow`, and hyper parameter optimization of the number of nodes and regularization parameter was done with a grid search on the variance unexplained.

### (v) Weighted linear combination of model

Given a set of $n$ interpolation models $\{f^1, f^2, \ldots f^n\}$ and a dataset $\mathcal{D}_t$, we can improve on the individual models by forming a weighted average

$$f_{\mathcal{D}_t} = \alpha_1 f^1_{\mathcal{D}_t} + \alpha_2 f^2_{\mathcal{D}_t} + \cdots + \alpha_n f^n_{\mathcal{D}_t}, \tag{3.19}$$

where $\alpha_1, \alpha_2, \ldots, \alpha_n$ are the weights. The weights are regarded as hyper parameters. If the number of models $n$ is not too high, there is little risk of overfitting, and the hyper parameters can simply be directly fitted to minimize the quality of fit. In §4, we use this method to combine the random forest and random Fourier features models.

## 4. Results

We begin the results section by establishing a suitable choice for the number of time samples $|\mathcal{K}|$, as discussed in §2c. We are looking to satisfy two main conditions. First, $\widetilde{\mathcal{Q}}$ needs to be approximately normally distributed. As seen from figure 4, the central limit theorem seems to hold for estimates of $\mathcal{E}$ with sample sizes $|\mathcal{K}| > 50$. Using normality of $\mathcal{Q}$ and $\mathcal{E}$ means that two standard deviations of $\mathcal{E}$ and $\mathcal{Q}$ correspond to 95% confidence intervals for $\mathcal{E}$ and $\mathcal{Q}$, respectively. Second, $|\mathcal{K}|$ needs to be sufficiently large for the error to be reasonably small. We chose $|\mathcal{K}| = 500$. Tables 1 and 2 show that this sample size achieves the goal of 1% relative error with 95% confidence for both $\mathcal{Q}$ and $\Delta\mathcal{Q}$ that we formulated in §2c. The only exception is the nearest neighbours model, for which the confidence interval for the error estimate is 2%. This is still sufficient accuracy for our purpose.

The quality of fit measurements reported in table 1 were all obtained using this sample size, and the reported uncertainty corresponds to two standard deviations, estimated according to (2.5). As the table shows, the confidence bounds vary slightly depending on the model. We observe that the distribution is narrower for better performing models. The same sample size was also used for the differences $\Delta\mathcal{Q}$ between the quality of fit of the benchmarking models and the random Fourier features model listed in table 2.

Figure 5 shows examples of the reconstructed wind field for the Fourier series, random Fourier features and Universal kriging interpolation models.[2] Figure 6 shows hourly means of the variance explained for different seasons throughout 2018.

The hyper parameter grid searches for the random Fourier features model are shown in figures 7 and 8. The optimal hyper parameters for the random Fourier features model were 0.01 for the Sobolev regularization constant $\lambda$, 0.001 for the divergence penalty $\eta$, 1.4 for the exponent $\gamma$ and 2.25 for the step size $\sigma$ in the proposal kernel in the adaptive Metropolis algorithm (algorithm 1). Running the Metropolis algorithm for $B = 500$ steps struck a good balance between convergence and computation time. Figure 9 shows an example of the frequency distribution that the algorithm samples from, and figure 10 shows the magnitude of the Fourier coefficients for the Fourier series model as a comparison.

---

[2]An animation of the reconstructed wind field for the random Fourier features model can be found at www.youtube.com/watch?v=eOMMJVPn0v8.

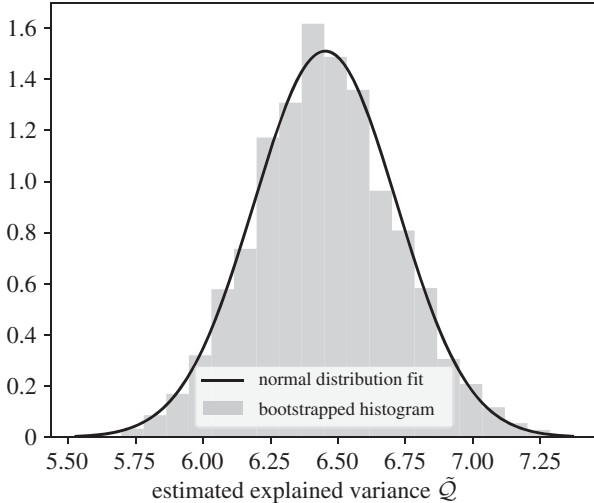

**Figure 4.** A histogram of 5000 samples of the variance unexplained $\tilde{\mathcal{Q}}(f) = \frac{1}{|\mathcal{K}|}\sum_{k \in \mathcal{K}} \tilde{\mathcal{Q}}_{t_k}(f)$ with $|\mathcal{K}| = 50$, bootstrapped from a total of 500 samples of $\tilde{\mathcal{Q}}_{t_k}$. The model $f$ is the Fourier series, and the black line is a fitted normal distribution.

**Table 1.** Quality of fit measurements $\mathcal{Q}$ and $\mathcal{E}$ with 95% confidence intervals for a number of different models. The dimension is indicated at the top row.

| interpolation model | $\widetilde{\mathcal{E}}$ [1] | $\widetilde{\mathcal{Q}}$ [m$^2$ s$^{-2}$] |
|---|---|---|
| nearest neighbours | $0.628 \pm 0.022$ | $11.145 \pm 0.386$ |
| inverse distance weighting | $0.407 \pm 0.014$ | $7.220 \pm 0.250$ |
| Universal kriging | $0.388 \pm 0.013$ | $6.887 \pm 0.235$ |
| random forest (RF) | $0.386 \pm 0.013$ | $6.862 \pm 0.238$ |
| neural network | $0.381 \pm 0.013$ | $6.762 \pm 0.225$ |
| Fourier series | $0.380 \pm 0.013$ | $6.740 \pm 0.226$ |
| random Fourier features (FF) | $0.370 \pm 0.012$ | $6.569 \pm 0.220$ |
| FF and RF average | $0.357 \pm 0.012$ | $6.333 \pm 0.212$ |

**Table 2.** Difference in quality of fit $\triangle\mathcal{Q}$ and $\triangle\mathcal{E}$ with 95% confidence intervals for the benchmark models relative to the random Fourier features model (FF). The dimension is indicated at the top row, in square brackets. A positive number means that the given model is worse than random Fourier features.

| interpolation model | $\triangle\widetilde{\mathcal{E}}$ [1] | $\triangle\widetilde{\mathcal{Q}}$ [m$^2$ s$^{-2}$] |
|---|---|---|
| nearest neighbours | $0.258 \pm 0.010$ | $4.576 \pm 0.183$ |
| inverse distance weighting | $0.037 \pm 0.003$ | $0.651 \pm 0.049$ |
| Universal kriging | $0.018 \pm 0.002$ | $0.318 \pm 0.033$ |
| random forest | $0.017 \pm 0.003$ | $0.293 \pm 0.060$ |
| neural network | $0.011 \pm 0.003$ | $0.192 \pm 0.056$ |
| Fourier series | $0.010 \pm 0.001$ | $0.171 \pm 0.017$ |
| FF and RF average | $-0.013 \pm 0.002$ | $-0.236 \pm 0.032$ |

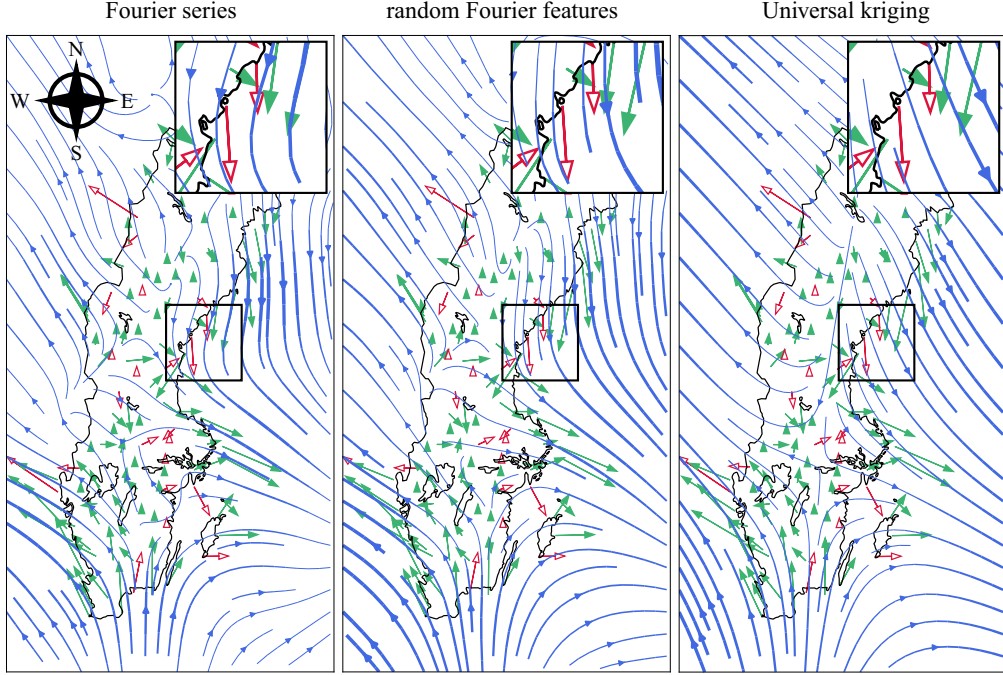

**Figure 5.** Reconstructions of the near-surface wind field in Sweden at 10.00, 22 January 2018. The blue streamlines show the reconstructed wind field, the green arrows show the wind velocity measurements at stations used for training the reconstruction, and the red hollow arrows are unseen test data points. From left to right, the fraction of variance unexplained for the above methods at this specific time is 0.57, 0.55 and 0.70. These values were measured on the test data points. (Online version in colour.)

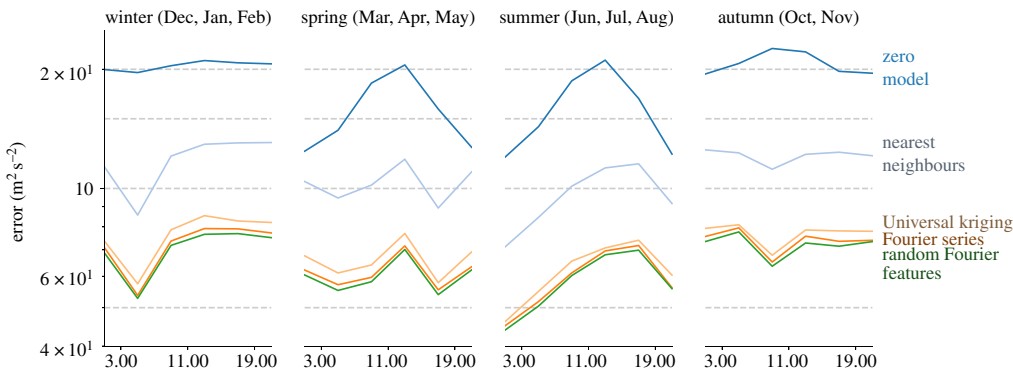

**Figure 6.** Hourly variance unexplained at different seasons, for a selection of the investigated interpolation models. The remaining models were left out to prevent clutter. The measurement is aggregated over all stations and plotted on a log-scale to improve visibility. Note that September is not in this data. The unexplained variance of the zero model corresponds to the mean square wind speed. (Online version in colour.)

We omit the hyper optimization results for the benchmarking models, which were all performed using the same type of grid search methods. The weights for the linear combination of the random forest and random Fourier features were chosen to minimize loss on the training data. We obtained a weight of 0.49 for the random forest, and 0.51 for random Fourier features.

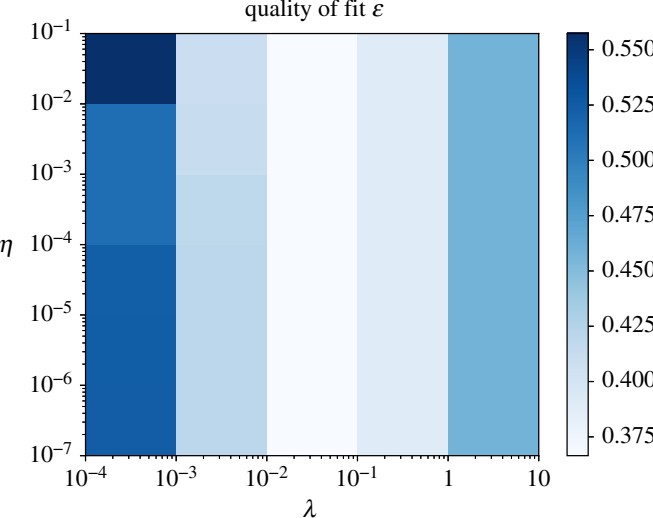

**Figure 7.** Fraction of variance unexplained (2.4) in the random Fourier features method, as a function of the Sobolev penalty $\lambda$ and divergence penalty $\eta$ explained in (3.4), with $\tau = 4 \times 10^7$, $\sigma = 2.25$, $\gamma = 1.25$. (Online version in colour.)

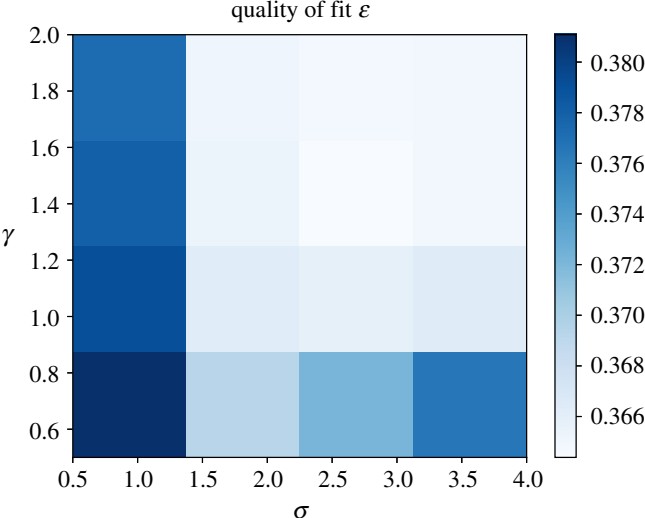

**Figure 8.** Fraction of variance unexplained (2.4) as a function of the exponent $\gamma$ and step size $\sigma$ in the adaptive Metropolis algorithm, with $\tau = 4 \times 10^7$, $\eta = 0.001$, $\lambda = 0.01$. (Online version in colour.)

## 5. Discussion

Table 1 shows that the model consisting of an average between the random forest and random Fourier features model performed the best out of the tested models. Table 2, consisting of the difference between quality of fit of the random Fourier features and the remaining models, indicates that this ordering is statistically significant, since none of the confidence intervals overlap with zero. In particular, the transition from the fixed frequencies in the Fourier series-based model to the randomly sampled frequencies of the random Fourier features results in a significant improvement. The enlarged area in figure 5 shows that the reconstructed wind field is

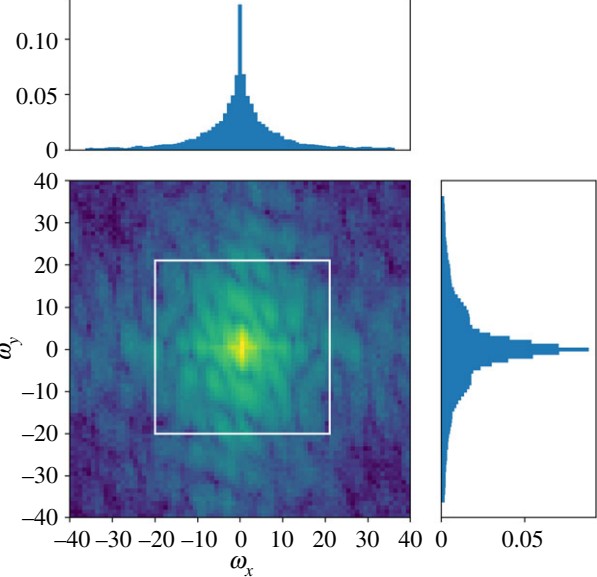

**Figure 9.** Sampling density $\rho(\omega) = \rho(\omega_x, \omega_y)$ of the random Fourier features algorithm for measurement data from 10.00, 22 January 2018. Algorithm (1) was run with $B = 1000$, collecting the Fourier frequencies at each step into a histogram. The profile plots show marginal distributions for the latitude frequencies $\omega_x$ (top) and longitude frequencies $\omega_y$ (right). (Online version in colour.)

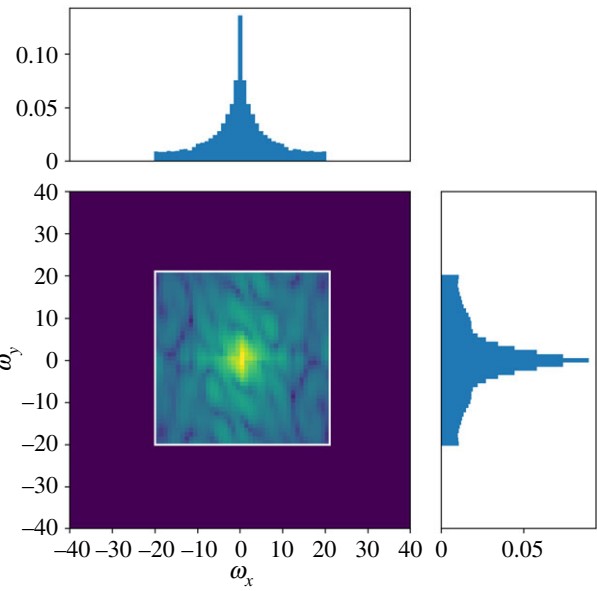

**Figure 10.** Heat map showing the magnitudes $||\widehat{\beta}||$ of the Fourier terms $\widehat{\beta}\, e^{i\omega \cdot x}$ as a function of the frequencies $\omega$ in the Fourier series model, for measurement data from 10.00, 22 January 2018. The support is a square grid with width 41, centred at the origin. The profile plots are analogous to figure 9. (Online version in colour.)

better aligned with unseen data when using random Fourier features. The numerical experiments in [15] indicate that the random Fourier features model outperforms its reference models as the amount of data increases. The improved performance is likely because the high-frequency details can be captured by exploring remote parts of the frequency domain, whereas a fixed grid of

frequencies centred at the origin cannot. Figure 9 shows a noticeable mass outside the white square, compared with figure 10, in which the high frequency contributions are truncated.

The success of the random forest and random Fourier features average hints that there might be more potential for improving the results using similar types of mixture models. It is also clear that the random Fourier features model on its own is competitive in comparison with the benchmarking models. A common argument for favouring statistical models such as kriging is easy access to precise error analysis, given that the prerequisite assumptions discussed in §3b(ii) hold. Whether or not this is worth trading in exchange for higher accuracy depends on the situation. The random Fourier features model has the upside of being easy to manipulate once it has been trained. It can be efficiently evaluated, differentiated and integrated for analysis of physical quantities such as divergence, vorticity and energy.

In figure 6, the relative performance of the interpolation models is visualized at different times of day and times of the year. The relative performance of the models is consistent throughout. A source of difficulty in wind field reconstruction is near-surface turbulence. In general, the interpolation error is lower during night-time. For the spring and summer months, this can be explained by the observed decrease in wind speeds during night-time. For autumn and winter, however, wind speeds are consistently high. The atmosphere tends to be more stable at night-time during the winter months [26], which might explain this decrease in error.

The autocorrelation plots in figures 2 and 3 demonstrate a strong autocorrelation in the wind field. Furthermore, figure 6 shows that the wind exhibits seasonal shifts in strength, as well as daily variations. This behaviour is well documented [26]. The models used in [12–14] are trained on multiple time samples whereas the spatial interpolation models used in our work only use the time aspect for hyper parameter optimization. Therefore, extending the spatial interpolation models to take time into account could improve the results. Note that since the data are only collected throughout 1 year, it is not possible to see the effect of inter-annual variability on the modelling error. More careful preprocessing of the data that account for these effects is needed to establish which model is optimal over a longer time interval.

As seen in figure 7, there is little to no change in the quality of fit when setting the divergence penalty to zero. Albeit a compelling idea, it makes sense that penalizing the divergence on a two-dimensional slice of the wind field would not improve the results. The near-surface wind flow is not typically parallel with the ground due to thermally and mechanically induced turbulence in the atmospheric boundary layer [27]. If we were to transition from a two-dimensional to a three-dimensional setting by incorporating elevation as a feature in the random Fourier features, divergence penalization would likely be more useful. More advanced methods can possibly incorporate no-slip or slipping boundary conditions on the ground surface, see [16].

Lastly, we discuss some general areas of improvement that are well documented and known in the environmental sciences community. First, the results may be improved by incorporating weather stations from nearby regions such as the Baltic Sea, Finland, Norway and Denmark. The random Fourier features model is constructed to efficiently find high frequency details in the target function, which makes it suitable when increasing the resolution of the data. The inclusion of additional covariates could also improve the results. Reinhardt & Samimi [3] include geopotential, potential vorticity, relative humidity, vertical velocity and elevation, and note that these covariates improve the results. Furthermore, orographic features like terrain roughness and topographic sheltering are known to affect the wind characteristics in Sweden [26].

## 6. Conclusion

In this report, we explored the potential for wind field reconstruction with sparse data using interpolation models. We investigated whether a novel interpolation model called random Fourier features model is competitive with respect to popular statistical interpolation models such as kriging, as well as modern machine learning methods such as random forests and neural networks. Drawing from the work [15], we derived an upper bound for the mean square error of the random Fourier Frequencies model, found a density that minimized this bound and devised

an adaptive Metropolis algorithm for sampling from this density. We showed that random Fourier features is competitive with respect to a time-space average of the square error and suggested some future areas of research, such as extending the model to incorporate data over multiple times and including more terrain-specific features. The key takeaway of our work is an improved spatial interpolation model for wind field reconstruction that is able to capture highly irregular near-surface wind features.

Data accessibility. Python code for this paper can be found on Bitbucket via the following url: https://bitbucket. org/the_eye_of_the_breeze/div-fourier-publication/src/master/. It implements interpolation methods discussed in §3 and contains the dataset used for training and evaluation.

Authors' contributions. J.K. designed and coordinated the study, was responsible for the data collection and pre-processing, designed the statistical framework, revised the manuscript and adapted the proof of Proposition A.1 to general regularization terms; E.S. carried out much of the statistical work, adapted the random Fourier algorithm to discrete frequencies and drafted the manuscript; R.T. conceived the study, is responsible for the theoretical motivation of the Fourier models, assisted in the proof of Proposition A.1 and revised the manuscript.

Competing interests. We declare we have no competing interests.

Funding. J.K. and R.T. were partially supported by the KAUST Office of Sponsored Research (OSR) under Award numbers OSR-2019-CRG8-4033 and OSR-2019-CRG8-4033.2, and the Alexander von Humboldt Foundation, through the Alexander von Humboldt Professorship award. E.S. was supported by the KAUST Visiting Student Research Program (VSRP).

Acknowledgements. The authors of this report would like to thank Prof. Anders Szepessy for his support and feedback. Furthermore, the work of Dmitry Kabanov, Luis Espath, Andreas Enblom and Magnus Tronstad in data processing and programming was integral in realizing the project. We also thank the two anonymous referees whose comments helped improve the manuscript.

# Appendix A

In this section, we derive an upper bound for the minimum of the expected loss $\mathbb{E}[\ell(\beta(x), u)]$ for the Fourier features model. We assume the standard regression setting where $u = f(x) + \epsilon$ and $\epsilon \in \mathbb{R}^2$ is independent of $x$, and $\mathbb{E}[||\epsilon||^2] = \sigma^2$. Let $\hat{f}(\omega)$ define the Fourier coefficients of $f$ and suppose for simplicity that $f$ is defined on the domain $X = [0, 2\pi] \times [0, 2\pi]$. That is, $f$ can be expressed as the Fourier series

$$f(x) = \frac{1}{2\pi} \sum_{\omega \in \mathbb{Z}^2} \hat{f}(\omega) \, e^{i\omega \cdot x}. \tag{A 1}$$

For the random Fourier features model, we choose

$$\beta(x) = \sum_{k=1}^{K} \widehat{\beta}_k \, e^{i\omega_k \cdot x}, \tag{A 2}$$

where $\widehat{\beta} = (\widehat{\beta}_1, \widehat{\beta}_2, \ldots, \widehat{\beta}_K)$ are the complex-valued two-dimensional coefficients and $\omega = (\omega_1, \omega_2, \ldots, \omega_K)$ are independent and identically distributed according to a discrete distribution $\rho : \mathbb{Z}^2 \to [0, \infty)$. The loss function is defined as

$$\ell(\beta(x), u) = ||\beta(x) - u||^2 + \lambda ||\mathcal{L}\beta||^2, \tag{A 3}$$

where

$$||\mathcal{L}\beta||^2 = \int_X \overline{\mathcal{L}\beta}(x) \mathcal{L}\beta(x) \, dx, \tag{A 4}$$

and $\mathcal{L} = \sum_{m=1}^{M} c_m \partial_1^{\alpha_{m,1}} \partial_2^{\alpha_{m,2}}$ is a linear differential operator with derivatives of at most order $d$ (i.e. $\alpha_{m,1} + \alpha_{m,2} \leq d$). Examples of regularizers that fit this description are the divergence penalty

$\mathcal{L}\beta = \nabla \cdot \beta = (\partial_1 + \partial_2)\beta$ (3.7), and each separate term of the Sobolev norm (3.5). The distribution $\rho$ is assumed to exist in a family $\mathcal{P}$ of discrete distributions

$$\mathcal{P} := \{\rho : \mathbb{Z}^2 \to (0,\infty) \big| \rho(\omega) > 0 \quad \text{and} \quad \mathbb{E}[|\omega_i|^m] < C, \quad 0 \le m \le 4d \ i = 1,2\}, \tag{A 5}$$

where $C$ is a positive real number. Thus, $\rho \in \mathcal{P}$ is strictly positive and has uniformly bounded moments $\sum_{\omega \in \mathbb{Z}^2} |\omega_i|^m \rho(\omega)$ of degree up to $4d$. Lastly, we make the assumption that

$$\frac{||\hat{f}(\omega)||}{\sum_{\omega' \in \mathbb{Z}^2} ||\hat{f}(\omega')||} \in \mathcal{P}, \tag{A 6}$$

which means that $f(x)$ has to be a member of the Sobolev space $W^{4d,2}(X)$.

**Proposition A.1.** *In the above setting, the following holds:*

(a) *The minimum of $\mathbb{E}[\ell(\beta(x), u)]$ with respect to the coefficients $\beta$ can be bounded:*

$$\mathbb{E}\left[\min_{\hat{\beta} \in \mathbb{C}^{2K}} \mathbb{E}[\ell(\beta(x), u) \mid \omega]\right] \le \frac{1 + \lambda\overline{C}}{(2\pi)^2 K} \sqrt{\mathbb{E}\left[\frac{||\hat{f}(\omega)||^4}{\rho(\omega)^4}\right]} + \sigma^2 - \frac{1}{K}\mathbb{E}[||f(x)||^2], \tag{A 7}$$

*where $\overline{C} > 0$.*

(b) *Furthermore, this upper bound is minimized by the distribution*

$$\rho(\omega) = \frac{||\hat{f}(\omega)||}{\sum_{\omega' \in \mathbb{Z}^2} ||\hat{f}(\omega')||}, \quad \omega \in \mathbb{Z}^2. \tag{A 8}$$

By replacing $\lambda\overline{C}$ in (A 7) with $\sum_{s=1}^{S} \lambda_s \overline{C}_s$, proposition A.1 can be generalized to include a linear combination of $S$ regularization functions $\sum_{s=1}^{S} \lambda_s ||\mathcal{L}_s \beta||^2$.

*Proof.* We divide the problem into part (a) and part (b) of the proposition.

(a) Let

$$\hat{\beta}_k = \frac{\hat{f}(\omega_k)}{2\pi K \rho(\omega_k)}, \quad k = 1, 2, \ldots, K. \tag{A 9}$$

$\beta$ is then a function of the random variables $x$ and $\omega$, which we denote $\beta(x; \omega)$. With this definition, $\beta$ is unbiased given $x$:

$$\mathbb{E}[\beta(x; \omega)|x] = \sum_{\omega \in \mathbb{Z}^2} K \frac{\hat{f}(\omega)}{2\pi K \rho(\omega)} e^{i\omega \cdot x} \rho(\omega) = \frac{1}{2\pi} \sum_{\omega' \in \mathbb{Z}^2} \hat{f}(\omega) e^{i\omega \cdot x} = f(x), \tag{A 10}$$

where we used that the components of $\omega$ are iid. Using independence of the residuals $\epsilon$ and the data $x$, we see that

$$\mathbb{E}[||\beta(x; \omega) - u||^2 \mid x]$$
$$= \mathbb{E}[||\beta(x; \omega) - (f(x) + \epsilon)||^2 \mid x] = \mathbb{E}[||\beta(x; \omega) - f(x)||^2 \mid x] + \sigma^2, \tag{A 11}$$

showing that the expected square error of $\beta$ as a function of $x$ is the variance of $\beta$ with respect to the frequencies $\omega$, plus the variance of the noise. The expected value of (A 11) can be simplified further since the frequencies $\omega_k$ are assumed independent

$$\mathbb{E}\left[\left|\left|\sum_{k=1}^{K} \hat{\beta}_k e^{i\omega_k \cdot x} - f(x)\right|\right|^2\right] \le \frac{1}{(2\pi)^2 K} \sqrt{\mathbb{E}\left[\frac{||\hat{f}(\omega)||^4}{\rho(\omega)^4}\right]} - \frac{1}{K}\mathbb{E}[||f(x)||^2]. \tag{A 12}$$

Here, $\omega$ denotes an arbitrary frequency with a distribution identical to the components of $\omega$. The last step is due to the Jensen inequality. Consider the penalty term containing the linear operator $\mathcal{L}$. Applying $\mathcal{L}$ to $\beta(x)$ is equivalent to multiplying each term $\hat{\beta}_{kj} e^{i\omega_k x}, j = 1, 2$ of the Fourier series with $\ell_j(\omega_k) = \sum_{m=1}^{M} c_m (i\omega_{k,1})^{\alpha_{m,1}} (i\omega_{k,2})^{\alpha_{m,2}}$, a

multivariate polynomial in $\omega_k$ of degree $d$. Define $r(\omega) = |\ell_1(\omega)|^2 + |\ell_2(\omega)|^2$. Note that $r$ has degree $2d$. It follows that

$$\mathbb{E}[||\mathcal{L}\beta||^2] = \int_X \mathbb{E}[|\mathcal{L}\beta(x;\omega)|^2 \big| x] dx$$
$$= K(2\pi)^2 \mathbb{E}[|\ell_1(\omega_k)\widehat{\beta}_{k1} + \ell_2(\omega_k)\widehat{\beta}_{k2}|^2]. \tag{A 13}$$

The first equality comes from switching the order of integration, and in the second we use independence of the Fourier frequencies combined with the size $(2\pi)^2$ of the region $X = [0, 2\pi] \times [0, 2\pi]$. Applying the Cauchy–Schwartz inequality twice on the last expression of (A 13) results in

$$K(2\pi)^2 \mathbb{E}[|\ell_1(\omega_k)\beta_{k1} + \ell_2(\omega_k)\beta_{k2}|^2] \leq K\mathbb{E}[r(\omega)||2\pi\beta||^2]$$

$$\leq \frac{1}{K}\sqrt{\mathbb{E}[r(\omega)^2]\mathbb{E}\left[\frac{||\hat{f}(\omega)||^4}{\rho(\omega)^4}\right]}. \tag{A 14}$$

The multivariate polynomial $r(\omega)^2 := \sum_{m=1}^{M'} r_m \omega_1^{\alpha_{m,1}} \omega_2^{\alpha_{m,2}}$ is of order $4d$, i.e. $\alpha_{m,1} + \alpha_{m,2} \leq 4d$. For each term $m$, define $\gamma_m := \alpha_{m,1}/\alpha_{m,1} + \alpha_{m,2} \in (0, 1)$. By assumption, $\rho$ lies in $\mathcal{P}$, and therefore, the expectation $\mathbb{E}[r(\omega)^2]$ can be uniformly bounded as follows:

$$\mathbb{E}[r(\omega)^2] = \sum_{m=1}^{M'} r_m \mathbb{E}[\omega_1^{\alpha_{m,1}} \omega_2^{\alpha_{m,2}}] \leq \sum_{m=1}^{M'} |r_m| \mathbb{E}[|\omega_1|^{(\alpha_{m,1}+\alpha_{m,2})\gamma_m} |\omega_2|^{(\alpha_{m,1}+\alpha_{m,2})(1-\gamma_m)}]. \tag{A 15}$$

Next, we used Hölder's inequality, which states that $\mathbb{E}[XY] \leq (\mathbb{E}[|X|^p])^{1/p}(\mathbb{E}[Y^q])^{1/q}$ for any $1/p + 1/q = 1$, term wise with $p = \frac{1}{\gamma_m}$.

$$\leq \sum_{m=1}^{M'} |r_m| \underbrace{(\mathbb{E}[|\omega_1|^{\alpha_{m,1}+\alpha_{m,2}}]^{\gamma_m} \mathbb{E}[|\omega_2|^{\alpha_{m,1}+\alpha_{m,2}}]^{1-\gamma_m}}_{\leq C^{\gamma_m} C^{1-\gamma_m} = C} \leq C \sum_{m=1}^{M'} |r_m|. \tag{A 16}$$

The last inequality comes from the bounded moments of the probability distribution $\rho$. If we define $\overline{C}$ such that $\frac{\overline{C}}{(2\pi)^2} = C \sum_{m=1}^{M'} |r_m|$ and put together (A 11), (A 12) and (A 16), we get

$$\mathbb{E}\left[\min_{\beta} \mathbb{E}[||\beta(x;\omega) - u||^2 + ||\mathcal{L}\beta||^2 | \omega]\right] \leq \frac{1 + \lambda\overline{C}}{(2\pi)^2 K}\sqrt{\mathbb{E}\left[\frac{||\hat{f}(\omega)||^4}{\rho(\omega)^4}\right]} - \frac{1}{K}\mathbb{E}[||f(x)||^2] + \sigma^2. \tag{A 17}$$

(b) To derive an optimal choice for $\rho$, we seek to minimize the expression inside the root in (A 17). We can redefine this problem in terms of a non-normalized function $p$ such that $\rho(\omega) = p(\omega)/\sum_{\mathbb{Z}^2} p(\omega')$:

$$\text{minimize} \quad \left(\sum_{\mathbb{Z}^2} \frac{||\hat{f}(\omega)||^4}{p(\omega)^3}\right) \cdot \left(\sum_{\mathbb{Z}^2} p(\omega)\right)^3. \tag{A 18}$$

First, define a real-valued function $H(\epsilon)$ where $\epsilon$ is a real number close to zero

$$H(\epsilon) = \left(\sum_{\mathbb{Z}^2} \frac{||\hat{f}(\omega)||^4}{(p(\omega) + \epsilon\delta(\omega))^3}\right) \cdot \left(\sum_{\mathbb{Z}^2} p(\omega) + \epsilon\delta(\omega)\right)^3, \tag{A 19}$$

and $\delta$ is a small arbitrary variation of $p$. Next, seek a solution $p$ to $H'(0) = 0$. After taking the derivative of (A 19) and simplifying, we are left with the following equation:

$$-c_1 \sum_{\mathbb{Z}^2} \frac{||\hat{f}(\omega)||^4}{p(\omega)^4}\delta(\omega) + c_2 \sum_{\mathbb{Z}^2} \delta(\omega) = 0, \tag{A 20}$$

where $c_1$ and $c_2$ are constants. The equation can be rewritten as

$$\sum_{\mathbb{Z}^2} \left( \frac{||\hat{f}(\omega)||^4}{p(\omega)^4} - \frac{c_2}{c_1} \right) \delta(\omega) = 0. \qquad (A\ 21)$$

Since $\delta(\omega)$ is arbitrary, the expression inside the sum must be zero, that is $p(\omega) = \sqrt[4]{c_2/c_1}||\hat{f}(\omega)||$ for all $\omega$. Hence, the optimal $\rho$ is $\rho(\omega) = ||\hat{f}(\omega)||/\sum_{\mathbb{Z}^2}||\hat{f}(\omega')||$. Lastly, some straightforward calculations show that the optimization with respect to $\rho$ is a convex optimization problem (that is, $\mathcal{P}$ is a convex set and $\mathbb{E}[||\hat{f}(\omega)||^4/\rho(\omega)^4]$ is a convex function with respect to $\rho$). But then, the above derived local minimum must also be a global minimum.

∎

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
