## [Peer Review File · Proceedings. Mathematical, Physical, and Engineering Sciences]

Review History

RSPA-2021-0236.R0 (Original submission)

Review form: Referee 1

Is the manuscript an original and important contribution to its field?

Good

Is the paper of sufficient general interest?

Good

Is the overall quality of the paper suitable?

Good

Can the paper be shortened without overall detriment to the main message?

Yes

Do you think some of the material would be more appropriate as an electronic appendix?

No

Do you have any ethical concerns with this paper?

Yes

Recommendation?

Major revision is needed (please make suggestions in comments)

Comments to the Author(s)

Summary

This study documents an approach to improve the description and characterization of near-ground wind fields. The approach is interesting and offers the potential to describe the characteristics of near-ground flow in an improved form than established interpolation approaches. From the authors' diction, it can be inferred that they define the problem as a mathematical one. This is ok, but it lacks the technical terms being standard in this field. It would be very desirable if the authors would also take the geophysical perspective because there have been technical terms (mainly from the field of meteorology) for the problem studied for decades, which I miss, and which are standard. Below I have listed comments on the manuscript that I believe will improve the quality of the manuscript.

Comments

P3L14: Reference [5] is old. There are now also approaches that model wind speed values directly on a grid, eliminating the need for interpolations.

P3L27: Please delete "(see Section (b))". This is redundant.

P3L29: Please specify "temperatures". Do you mean "air temperature"? There are about 30 kinds of "temperature" known in geophysics.

P3L31: Please lowercase "Kriging".

P3L34: Please delete "terrain" in front of elevation. Elevation is terrain height above sea level.

P4L8: The term "wind field reconstruction" has already been mentioned on P3L21. The repeated definition here is redundant and should be deleted.

P4L17: Do you mean station height above sea level when using "altitude"? If so, then please replace "altitude" by "elevation". Altitude is height above ground level, i.e., the measuring height of 10 m.

P4L19-P4L20: The angle of the horizontal wind vector is the wind direction, the direction from which the wind comes. 90° correspond to east (positive x-axis), 180° are south (negative y-axis), 270° are west (negative x-axis), and 360° are north (positive y-axis). 0° represent calms: no speed, no direction. Wind direction is measured clockwise relative to north. Wind speed is the absolute value of the horizontal 2D wind vector. At meteorological stations, the wind vector is mostly measured in the horizontal. I suggest replacing "velocity" / "wind velocity" with "wind speed" in the entire manuscript where appropriate because this is the technical term used in meteorology.

P4L23: Why is Figure 5 mentioned before Figs. 2-4 in this line. Please keep the order of referencing display items in the text.

P4L23: Please replace "velocities" by "wind speed".

P4L29-P4L30: Why did you remove wind measurements of September 2018? How many stations in the network showed "unusually" high wind speed? What is "unusually" in m/s? Was the entire network measuring wrong? Please elaborate.

P4L49-P4L50: Please delete the last sentence of this section. It is redundant.

P5L7, Figure 1:

- It is common to include a north arrow in maps being display items in studies with earth system context.
- I suggest using a color scheme that represents elevation, not altitude, of the weather stations. This enhances the perceptibility of the stations' distribution as a function of elevation compared to differently sized blue dots.

P5L38, P5L51, P5L57: Equations on P5 are not numbered. Please change. Please number all equations in the text.

P5L46-P5L47: The sentence in these lines seems to be incomplete. The subject is missing.

P6L18: The data samples are not independent, neither in space nor in time. You demonstrate this yourself in Figure 5. Sweden's territory as a study area is too small for independent wind speed values, dependent on the temporal scale.

P6L35-P6L37: Wind speed not only shows a strong annual cycle (max in winter (DJF) and min in summer (JJA)) in Sweden but also a diurnal cycle, and a dependence on the passage of multi-day weather systems (high- and low-pressure areas). Besides, wind speed shows inter-annual variability and a trend component. Therefore, the use of only one year of data cannot be used to map the entire wind vector dynamics. What component of the measured wind signals did you simulate? The measured signal itself or did you account for diurnal cycle, annual cycle, ...?

P8L41: What is "Algorithm 1"? Please specify.

P8L42: Please uppercase "Fourier".

P9L38: Please delete "for short" and lowercase "Kriging".

P9L46: What is the definition of "weighted average of the velocity measurements" here. Is this the wind speed? Or is it the horizontal wind vector components?

P10L7: Do you mean the measuring heights with "altitudes"? Or do you mean "elevations" corresponding to the height above sea level? One of the most severe issues in interpolating wind speed and wind direction is the treatment of complex terrain where wind speed and wind direction cannot be interpolated between weather stations that mostly measure near settlements in flat and smooth (without surface roughness, i.e., grassland) terrain. Any kind of wind speed and wind direction interpolation method must fail to some extent.

P10L37: Please delete "for short" and lowercase "kriging". Here, and everywhere else in the text where appropriate.

P10L56: Please replace "altitude" with "elevation".

P11L54: Please lowercase "Where".

P12L37: Please uppercase "table".

P12L46: What is "Algorithm 1"? Please specify.

P12L53-P12L56: Tables 1 and 2 summarize results. It should thus be part of the results and not of the discussion section.

P14L7, Figure 7: Why is the displayed fraction of unexplained variance sometimes greater than 1? Please clarify.

P14L55-P14L56: Does it make sense to compare wind data measured in a meteorological network with fluid flow around a cylinder? This is something completely different. Wind speed and wind direction measured in a meteorological network is at least subject to orography (topographic sheltering, relative elevation, absolute elevation, ...), surface roughness (forested areas, urban areas, agricultural areas, ...), sequence of highs and lows, diurnal cycle, annual cycle, inter-annual variability, and trend while flow around a cylinder is not.

P14L58-P15L30: Please delete “[7, p. 18]”. This is redundant.

P14-P16: Are Figures 7-10 referenced in the text? I found no reference. Why are they presented then? Please include the figures in the text or delete them.

P15L7, Figure 10: This is the first time I read “trajectory”. What is the meaning of this figure? Does the number of trajectories correspond to the number of frequencies which was stated to be 400? Should P15L23 read “The remaining 399 trajectories”?

P15L43: I not aware of any study that uses measured wind data from meteorological stations in which air pressure, air temperature or air humidity did improve the wind field modeling. Please provide a reference or delete these meteorological variables.

P15L38-P15L46: These findings are well known for decades and included in many previous wind field modeling studies.

P15L49: Air flow is seldom parallel to the ground. This is because in the atmospheric boundary layer it is mainly turbulent, at least during daytime (thermally induced turbulence) and above rough surfaces (mechanically induced turbulence). Atmospheric stability which changes over the day and the year determines flow characteristics.

P16L7-P16L15: “7. Conclusion” is no conclusion it is pure summary. Please adapt contents of this section to the heading and draw conclusions for your work.

P16L15: Could you provide a map with the interpolated wind field in the study area? This would be extremely helpful in judging the useability of the presented approach, especially in complex terrain.

Review form: Referee 2

Is the manuscript an original and important contribution to its field?

Good

Is the paper of sufficient general interest?

Acceptable

Is the overall quality of the paper suitable?

Good

Can the paper be shortened without overall detriment to the main message?

Yes

Do you think some of the material would be more appropriate as an electronic appendix?

No

Do you have any ethical concerns with this paper?

No

Recommendation?

Major revision is needed (please make suggestions in comments)

Comments to the Author(s)

In this manuscript, the authors demonstrate a method for reconstructing a two-dimensional wind field from sparse measurements, specifically adaptive random Fourier features with Metropolis sampling, which is based on previous work by some of the authors. This method is applied to measurements from weather stations throughout Sweden to assess how accurately the two-dimensional wind vectors can be reconstructed at different points. The accuracy is compared to results from several other interpolation models which are commonly used for such applications. Overall, the adaptive random Fourier features method performs slightly better than the highest performing reference methods, demonstrating its usefulness for wind field reconstruction problems.

Overall, the manuscript is written very clearly, although there are some points that could use clarification. The authors also discuss interesting extensions of the algorithm that could further improve its performance for wind field reconstruction applications. There are a few major areas where the manuscript should be improved before it is accepted, however. These are listed below followed by many smaller comments.

First, the introduction motivates the wind field reconstruction problem by mentioning wind farm planning and weather forecasting applications, where it is written that "The available data is often sparse and interpolation techniques are employed in order to increase spatial resolution." However, the paragraph includes only a couple references and doesn't provide specific examples for why spatial interpolation is needed. More detailed motivation and references should be provided in the introduction. In wind farm planning, for example, a meteorological mast is typically installed at the site of interest where a measurement campaign is performed to determine the potential of the site, rather than spatially interpolate from surrounding measurement locations. For example, see:

Clifton, et al. "Wind plant preconstruction energy estimates: Current practice and opportunities," Technical report, NREL/TP-5000-64735, National Renewable Energy Laboratory, Golden, CO, 2016.

Additionally, it is unclear from the introduction how spatial interpolation would help with weather forecasting.

Second, a reference is provided for the derivation of the upper bound equation, which much of the algorithm relies on, on page 7, ln. 30. But the reference [14] appears to be the current paper and only the authors, title, and year are provided. Since this equation is a fundamental part of the manuscript, the derivation should either be provided in the manuscript, or a peer-reviewed publication containing the derivation should be cited.

Next, Figures 7-10 do not contain any reference or discussion in the body of the paper. They should be discussed in the body of the text, otherwise it is unclear what purpose they serve.

Next, to illustrate the results of the wind field construction algorithm, and help make the paper appeal to a wide readership, a plot showing the reconstructed wind field across Sweden for a particular time sample would be very informative.

Another comment is that throughout the manuscript, I was wondering why time was not included as a feature in the wind field reconstruction algorithm. For example, the past wind speeds at points upstream of the location of interest could provide useful information about the

current wind speeds after the flow advects downstream. I was happy to see that this was brought up in the Discussion section, but it might be helpful to acknowledge that time could be another important feature earlier in the paper and possibly explain why it was not considered.

The last major comment is that it would be interesting to look at the dependence of the wind field reconstruction accuracy on the atmospheric conditions, season, time of day, etc. For example, in daytime in the summer, when there is more turbulence due to surface heating, or nighttime in winter when the atmosphere is expected to be more stable. Can some discussion be added on this topic? The time period where the measurements are obtained for the results should be mentioned as well.

Other comments:

1. The abstract contains references to other work [13] and [19]. Abstracts are typically meant to serve as standalone descriptions of the work, so the references should be removed (unless this is specifically allowed by the journal).
2. Page 2, ln. 17: "interpolating wind over shorter time intervals" Be specific about what you mean by "shorter." There is no previous time interval mentioned.
3. Page 2, ln. 56: "Typically, air flow is assumed to satisfy the incompressible..." A reference should be provided.
4. Page 3, ln. 18: "Each measurement is a 10-minute average" Above, it is stated that hourly wind velocities are provided. Can you clarify this? Is a 10-minute average measurement provided once per hour?
5. Page 3, ln. 29: "...removed because of unusually high wind speeds." What is the motivation for removing the period with high wind speeds?
6. Page 4, ln. 41 "Even if the number of measurements was sufficient, model training is often cumbersome." This is a little confusing, because you are still performing model training in your algorithm.
7. Page 4, ln. 52: In the equation, can you explain why the subscript in $\tilde{Q}_k(f)$ is "k" rather than " t_k " like it is for " $Q(f)$ "?
8. Figure 1: A legend for the altitude indicated by the diameters as well as axis ticks and labels would be helpful. Additionally, the region showing the size of the 2D period used in the Fourier series approximation would be very informative.
9. Page 6, ln. 22: Provide a reference for Sobolev-norm.
10. Page 6, ln. 55: "We went with $M = 10$ " Explain why you chose this value.
11. Page 7, ln. 17: "In this way, the random Fourier features is similar to a fully connected neural network with one hidden layer..." Please provide more explanation about this, or a reference, since this is not obvious for the general reader.
12. Page 7, ln. 30: \overline{C} needs to be explained. Further, why is ω on the left hand side bold (indicating a vector) but not bold on the right hand side? Are they different variables? Additionally, the Metropolis algorithm is used to minimize the upper bound. Can you clarify whether this causes the left hand side and right hand side to become equal, or does minimizing the RHS simply reduce the upper bound without any guarantees about minimizing the expected loss on the LHS? Lastly, are there any guarantees about how closely the solution approximates the solution to the actual minimization problem of interest on Page 7, ln. 15?
13. Page 7, ln. 41: "the weights ω ..." Why is ω referred to as weights here, instead of frequencies?
14. Page 7, ln. 43: σ has not been defined yet.
15. Page 7, ln. 47: "the optimal choice for γ as..." Explain in what sense it is optimal?
16. Page 7, ln. 48: "the optimal variance of the proposed kernel..." The variance at the end of the sentence is written as σ . Should this be σ^2 ?
17. Section 4 (b): References for Kriging and feedforward neural networks should be provided in those sections.
18. Page 9, ln. 44: "with a linear variogram" As described, the Kriging algorithm also requires a covariogram. Explain what type of covariogram is used.
19. Page 9, ln. 48: Some basic explanation of regression trees would be appreciated. This

section seems to assume the reader is already familiar with regression trees and random forests.

20. Page 9, ln. 55: "trained on a polynomial feature map" Explain why a polynomial feature map is used. Is this a common choice for random forests? Additionally, the definition of the polynomial features contains the term $1^{\{p_1\}}$. Won't this always be one?

21. Section 5: Please explain the relative weights for the random Fourier features and random forest models for the "FF and RF average" method. Are they weighted equally?

22. Page 11, ln. 36: "We went with $|K| = 500$ " As mentioned earlier, please provide some information about when the samples occur. Or are they randomly sampled from the data set?

23. Figs. 8 and 9: As described in Section 4 (a) (ii), the optimal $\rho(\omega)$ is proportional to $|\beta(\omega)|$. Can you discuss in the text how closely the distributions in Fig. 8 and Fig. 9 match and the implications for the optimality of the solution from the Metropolis algorithm?

24. Page 13, ln. 56: "The main difference is that our data is extremely sparse..." Another difference could be that the local influence of terrain exacerbates the sparse data issue.

25. Page 14, ln. 34: Please explain how Figure 5 shows clear seasonality. The different seasons are not indicated in the plot as far as I can tell.

26. Page 14, ln. 53: "An alternative approach is to model the wind flow in two layers..." As an example, what might the two layers be?

27. Page 15, ln. 15: It seems more appropriate to say that you provided some "possible" way to improve the results, or something similar.

Minor comments:

1. Page 5, ln. 32: "The equations (3.2) and (3.3) were used in Section 5..." Section 5 has not been introduced yet, so "were used" does not seem appropriate.

2. Page 5, ln. 55: "number terms" -> "number of terms"

3. Page 7, ln. 25: "same the" -> "the same"

4. Page 7, ln. 55: "insure" -> "ensure"?

5. Page 8, ln. 41: "in Section (b)" Here and elsewhere throughout the paper the main section number is missing in the reference.

6. Page 9, ln. 54: "in scikit-learn" It should be mentioned that this is a python package.

Decision letter (RSPA-2021-0236.R0)

27-Jul-2021

Dear Dr Ström

The Editor of Proceedings A has now received comments from referees on the above paper and would like you to revise it in accordance with their suggestions which can be found below (not including confidential reports to the Editor).

Please submit a copy of your revised paper within four weeks - if we do not hear from you within this time then it will be assumed that the paper has been withdrawn. In exceptional circumstances, extensions may be possible if agreed with the Editorial Office in advance.

Please note that it is the editorial policy of Proceedings A to offer authors one round of revision in which to address changes requested by referees. If the revisions are not considered satisfactory by the Editor, then the paper will be rejected, and not considered further for publication by the journal. In the event that the author chooses not to address a referee's comments, and no scientific justification is included in their cover letter for this omission, it is at the discretion of the Editor whether to continue considering the manuscript.

To revise your manuscript, log into <http://mc.manuscriptcentral.com/prsa> and enter your Author Centre, where you will find your manuscript title listed under "Manuscripts with

Decisions." Under "Actions," click on "Create a Revision." Your manuscript number has been appended to denote a revision.

You will be unable to make your revisions on the originally submitted version of the manuscript. Instead, revise your manuscript and upload a new version through your Author Centre.

When submitting your revised manuscript, you will be able to respond to the comments made by the referee(s) and upload a file "Response to Referees" in Step 1: "View and Respond to Decision Letter". Please provide a point-by-point response to the comments raised by the reviewers and the editor(s). A thorough response to these points will help us to assess your revision quickly. You can also upload a 'tracked changes' version either as part of the 'Response to reviews' or as a 'Main document'.

IMPORTANT: Your original files are available to you when you upload your revised manuscript. Please delete any unnecessary previous files before uploading your revised version.

When revising your paper please ensure that it remains under 28 pages long. In addition, any pages over 20 will be subject to a charge (£150 + VAT (where applicable) per page). Your paper has been ESTIMATED to be 16 pages.

Open Access

You are invited to opt for open access, our author pays publishing model. Payment of open access fees will enable your article to be made freely available via the Royal Society website as soon as it is ready for publication. For more information about open access please visit <https://royalsociety.org/journals/authors/open-access/>. The open access fee for this journal is £1700/\$2380/€2040 per article. VAT will be charged where applicable. Please note that if the corresponding author is at an institution that is part of a Read and Publishing deal you are required to select this option. See <https://royalsociety.org/journals/librarians/purchasing/read-and-publish/read-publish-agreements/> for further details.

Once again, thank you for submitting your manuscript to Proc. R. Soc. A and I look forward to receiving your revision. If you have any questions at all, please do not hesitate to get in touch.

Yours sincerely
Raminder Shergill
proceedingsa@royalsociety.org

on behalf of
Professor Johannes Zimmer
Board Member
Proceedings A

Reviewer(s)' Comments to Author:

Referee: 1

Comments to the Author(s)

Summary

This study documents an approach to improve the description and characterization of near-ground wind fields. The approach is interesting and offers the potential to describe the characteristics of near-ground flow in an improved form than established interpolation approaches. From the authors' diction, it can be inferred that they define the problem as a mathematical one. This is ok, but it lacks the technical terms being standard in this field. It would be very desirable if the authors would also take the geophysical perspective because there have been technical terms (mainly from the field of meteorology) for the problem studied for decades, which I miss, and which are standard. Below I have listed comments on the manuscript that I believe will improve the quality of the manuscript.

Comments

P3L14: Reference [5] is old. There are now also approaches that model wind speed values directly on a grid, eliminating the need for interpolations.

P3L27: Please delete “(see Section (b))”. This is redundant.

P3L29: Please specify “temperatures”. Do you mean “air temperature”? There are about 30 kinds of “temperature” known in geophysics.

P3L31: Please lowercase “Kriging”.

P3L34: Please delete “terrain” in front of elevation. Elevation is terrain height above sea level.

P4L8: The term “wind field reconstruction” has already been mentioned on P3L21. The repeated definition here is redundant and should be deleted.

P4L17: Do you mean station height above sea level when using “altitude”? If so, then please replace “altitude” by “elevation”. Altitude is height above ground level, i.e., the measuring height of 10 m.

P4L19-P4L20: The angle of the horizontal wind vector is the wind direction, the direction from which the wind comes. 90° correspond to east (positive x-axis), 180° are south (negative y-axis), 270° are west (negative x-axis), and 360° are north (positive y-axis). 0° represent calms: no speed, no direction. Wind direction is measured clockwise relative to north. Wind speed is the absolute value of the horizontal 2D wind vector. At meteorological stations, the wind vector is mostly measured in the horizontal. I suggest replacing “velocity” / “wind velocity” with “wind speed” in the entire manuscript where appropriate because this is the technical term used in meteorology.

P4L23: Why is Figure 5 mentioned before Figs. 2-4 in this line. Please keep the order of referencing display items in the text.

P4L23: Please replace “velocities” by “wind speed”.

P4L29-P4L30: Why did you remove wind measurements of September 2018? How many stations in the network showed “unusually” high wind speed? What is “unusually” in m/s? Was the entire network measuring wrong? Please elaborate.

P4L49-P4L50: Please delete the last sentence of this section. It is redundant.

P5L7, Figure 1:

- It is common to include a north arrow in maps being display items in studies with earth system context.

- I suggest using a color scheme that represents elevation, not altitude, of the weather stations. This enhances the perceptibility of the stations’ distribution as a function of elevation compared to differently sized blue dots.

P5L38, P5L51, P5L57: Equations on P5 are not numbered. Please change. Please number all equations in the text.

P5L46-P5L47: The sentence in these lines seems to be incomplete. The subject is missing.

P6L18: The data samples are not independent, neither in space nor in time. You demonstrate this yourself in Figure 5. Sweden’s territory as a study area is too small for independent wind speed values, dependent on the temporal scale.

P6L35-P6L37: Wind speed not only shows a strong annual cycle (max in winter (DJF) and min in summer (JJA)) in Sweden but also a diurnal cycle, and a dependence on the passage of multi-day weather systems (high- and low-pressure areas). Besides, wind speed shows inter-annual variability and a trend component. Therefore, the use of only one year of data cannot be used to map the entire wind vector dynamics. What component of the measured wind signals did you simulate? The measured signal itself or did you account for diurnal cycle, annual cycle, ...?

P8L41: What is "Algorithm 1"? Please specify.

P8L42: Please uppercase "Fourier".

P9L38: Please delete "for short" and lowercase "Kriging".

P9L46: What is the definition of "weighted average of the velocity measurements" here. Is this the wind speed? Or is it the horizontal wind vector components?

P10L7: Do you mean the measuring heights with "altitudes"? Or do you mean "elevations" corresponding to the height above sea level? One of the most severe issues in interpolating wind speed and wind direction is the treatment of complex terrain where wind speed and wind direction cannot be interpolated between weather stations that mostly measure near settlements in flat and smooth (without surface roughness, i.e., grassland) terrain. Any kind of wind speed and wind direction interpolation method must fail to some extent.

P10L37: Please delete "for short" and lowercase "kriging". Here, and everywhere else in the text where appropriate.

P10L56: Please replace "altitude" with "elevation".

P11L54: Please lowercase "Where".

P12L37: Please uppercase "table".

P12L46: What is "Algorithm 1"? Please specify.

P12L53-P12L56: Tables 1 and 2 summarize results. It should thus be part of the results and not of the discussion section.

P14L7, Figure 7: Why is the displayed fraction of unexplained variance sometimes greater than 1? Please clarify.

P14L55-P14L56: Does it make sense to compare wind data measured in a meteorological network with fluid flow around a cylinder? This is something completely different. Wind speed and wind direction measured in a meteorological network is at least subject to orography (topographic sheltering, relative elevation, absolute elevation, ...), surface roughness (forested areas, urban areas, agricultural areas, ...), sequence of highs and lows, diurnal cycle, annual cycle, inter-annual variability, and trend while flow around a cylinder is not.

P14L58-P15L30: Please delete "[7, p. 18]". This is redundant.

P14-P16: Are Figures 7-10 referenced in the text? I found no reference. Why are they presented then? Please include the figures in the text or delete them.

P15L7, Figure 10: This is the first time I read "trajectory". What is the meaning of this figure? Does the number of trajectories correspond to the number of frequencies which was stated to be 400? Should P15L23 read "The remaining 399 trajectories"?

P15L43: I not aware of any study that uses measured wind data from meteorological stations in which air pressure, air temperature or air humidity did improve the wind field modeling. Please provide a reference or delete these meteorological variables.

P15L38-P15L46: These findings are well known for decades and included in many previous wind field modeling studies.

P15L49: Air flow is seldom parallel to the ground. This is because in the atmospheric boundary layer it is mainly turbulent, at least during daytime (thermally induced turbulence) and above rough surfaces (mechanically induced turbulence). Atmospheric stability which changes over the day and the year determines flow characteristics.

P16L7-P16L15: "7. Conclusion" is no conclusion it is pure summary. Please adapt contents of this section to the heading and draw conclusions for your work.

P16L15: Could you provide a map with the interpolated wind field in the study area? This would be extremely helpful in judging the useability of the presented approach, especially in complex terrain.

Referee: 2

Comments to the Author(s)

In this manuscript, the authors demonstrate a method for reconstructing a two-dimensional wind field from sparse measurements, specifically adaptive random Fourier features with Metropolis sampling, which is based on previous work by some of the authors. This method is applied to measurements from weather stations throughout Sweden to assess how accurately the two-dimensional wind vectors can be reconstructed at different points. The accuracy is compared to results from several other interpolation models which are commonly used for such applications. Overall, the adaptive random Fourier features method performs slightly better than the highest performing reference methods, demonstrating its usefulness for wind field reconstruction problems.

Overall, the manuscript is written very clearly, although there are some points that could use clarification. The authors also discuss interesting extensions of the algorithm that could further improve its performance for wind field reconstruction applications. There are a few major areas where the manuscript should be improved before it is accepted, however. These are listed below followed by many smaller comments.

First, the introduction motivates the wind field reconstruction problem by mentioning wind farm planning and weather forecasting applications, where it is written that "The available data is often sparse and interpolation techniques are employed in order to increase spatial resolution." However, the paragraph includes only a couple references and doesn't provide specific examples for why spatial interpolation is needed. More detailed motivation and references should be provided in the introduction. In wind farm planning, for example, a meteorological mast is typically installed at the site of interest where a measurement campaign is performed to determine the potential of the site, rather than spatially interpolate from surrounding measurement locations. For example, see:

Clifton, et al. "Wind plant preconstruction energy estimates: Current practice and opportunities," Technical report, NREL/TP-5000-64735, National Renewable Energy Laboratory, Golden, CO, 2016.

Additionally, it is unclear from the introduction how spatial interpolation would help with weather forecasting.

Second, a reference is provided for the derivation of the upper bound equation, which much of the algorithm relies on, on page 7, ln. 30. But the reference [14] appears to be the current paper and only the authors, title, and year are provided. Since this equation is a fundamental part of the manuscript, the derivation should either be provided in the manuscript, or a peer-reviewed publication containing the derivation should be cited.

Next, Figures 7-10 do not contain any reference or discussion in the body of the paper. They should be discussed in the body of the text, otherwise it is unclear what purpose they serve.

Next, to illustrate the results of the wind field construction algorithm, and help make the paper appeal to a wide readership, a plot showing the reconstructed wind field across Sweden for a particular time sample would be very informative.

Another comment is that throughout the manuscript, I was wondering why time was not included as a feature in the wind field reconstruction algorithm. For example, the past wind speeds at points upstream of the location of interest could provide useful information about the current wind speeds after the flow advects downstream. I was happy to see that this was brought up in the Discussion section, but it might be helpful to acknowledge that time could be another important feature earlier in the paper and possibly explain why it was not considered.

The last major comment is that it would be interesting to look at the dependence of the wind field reconstruction accuracy on the atmospheric conditions, season, time of day, etc. For example, in daytime in the summer, when there is more turbulence due to surface heating, or nighttime in winter when the atmosphere is expected to be more stable. Can some discussion be added on this topic? The time period where the measurements are obtained for the results should be mentioned as well.

Other comments:

1. The abstract contains references to other work [13] and [19]. Abstracts are typically meant to serve as standalone descriptions of the work, so the references should be removed (unless this is specifically allowed by the journal).
2. Page 2, ln. 17: "interpolating wind over shorter time intervals" Be specific about what you mean by "shorter." There is no previous time interval mentioned.
3. Page 2, ln. 56: "Typically, air flow is assumed to satisfy the incompressible..." A reference should be provided.
4. Page 3, ln. 18: "Each measurement is a 10-minute average" Above, it is stated that hourly wind velocities are provided. Can you clarify this? Is a 10-minute average measurement provided once per hour?
5. Page 3, ln. 29: "...removed because of unusually high wind speeds." What is the motivation for removing the period with high wind speeds?
6. Page 4, ln. 41 "Even if the number of measurements was sufficient, model training is often cumbersome." This is a little confusing, because you are still performing model training in your algorithm.
7. Page 4, ln. 52: In the equation, can you explain why the subscript in $\tilde{Q}_k(f)$ is "k" rather than "t_k" like it is for "Q(f)"?
8. Figure 1: A legend for the altitude indicated by the diameters as well as axis ticks and labels would be helpful. Additionally, the region showing the size of the 2D period used in the Fourier series approximation would be very informative.
9. Page 6, ln. 22: Provide a reference for Sobolev-norm.
10. Page 6, ln. 55: "We went with $M = 10$ " Explain why you chose this value.
11. Page 7, ln. 17: "In this way, the random Fourier features is similar to a fully connected neural network with one hidden layer..." Please provide more explanation about this, or a reference, since this is not obvious for the general reader.
12. Page 7, ln. 30: \overline{C} needs to be explained. Further, why is ω on the left hand side bold (indicating a vector) but not bold on the right hand side? Are they different variables? Additionally, the Metropolis algorithm is used to minimize the upper bound. Can you clarify

whether this causes the left hand side and right hand side to become equal, or does minimizing the RHS simply reduce the upper bound without any guarantees about minimizing the expected loss on the LHS? Lastly, are there any guarantees about how closely the solution approximates the solution to the actual minimization problem of interest on Page 7, ln. 15?

13. Page 7, ln. 41: "the weights ω ..." Why is ω referred to as weights here, instead of frequencies?

14. Page 7, ln. 43: σ has not been defined yet.

15. Page 7, ln. 47: "the optimal choice for γ as..." Explain in what sense it is optimal?

16. Page 7, ln. 48: "the optimal variance of the proposed kernel..." The variance at the end of the sentence is written as σ . Should this be σ^2 ?

17. Section 4 (b): References for Kriging and feedforward neural networks should be provided in those sections.

18. Page 9, ln. 44: "with a linear variogram" As described, the Kriging algorithm also requires a covariogram. Explain what type of covariogram is used.

19. Page 9, ln. 48: Some basic explanation of regression trees would be appreciated. This section seems to assume the reader is already familiar with regression trees and random forests.

20. Page 9, ln. 55: "trained on a polynomial feature map" Explain why a polynomial feature map is used. Is this a common choice for random forests? Additionally, the definition of the polynomial features contains the term 1^{p_1} . Won't this always be one?

21. Section 5: Please explain the relative weights for the random Fourier features and random forest models for the "FF and RF average" method. Are they weighted equally?

22. Page 11, ln. 36: "We went with $|K| = 500$ " As mentioned earlier, please provide some information about when the samples occur. Or are they randomly sampled from the data set?

23. Figs. 8 and 9: As described in Section 4 (a) (ii), the optimal $\rho(\omega)$ is proportional to $|\beta(\omega)|$. Can you discuss in the text how closely the distributions in Fig. 8 and Fig. 9 match and the implications for the optimality of the solution from the Metropolis algorithm?

24. Page 13, ln. 56: "The main difference is that our data is extremely sparse..." Another difference could be that the local influence of terrain exacerbates the sparse data issue.

25. Page 14, ln. 34: Please explain how Figure 5 shows clear seasonality. The different seasons are not indicated in the plot as far as I can tell.

26. Page 14, ln. 53: "An alternative approach is to model the wind flow in two layers..." As an example, what might the two layers be?

27. Page 15, ln. 15: It seems more appropriate to say that you provided some "possible" way to improve the results, or something similar.

Minor comments:

1. Page 5, ln. 32: "The equations (3.2) and (3.3) were used in Section 5..." Section 5 has not been introduced yet, so "were used" does not seem appropriate.

2. Page 5, ln. 55: "number terms" -> "number of terms"

3. Page 7, ln. 25: "same the" -> "the same"

4. Page 7, ln. 55: "insure" -> "ensure"?

5. Page 8, ln. 41: "in Section (b)" Here and elsewhere throughout the paper the main section number is missing in the reference.

6. Page 9, ln. 54: "in scikit-learn" It should be mentioned that this is a python package.

Comment from subject editor: Kriging was named after D. G. Krige so the capital letter is still an option.

Author's Response to Decision Letter for (RSPA-2021-0236.R0)

See Appendix A.

RSPA-2021-0236.R1 (Revision)

Review form: Referee 1

Is the manuscript an original and important contribution to its field?

Good

Is the paper of sufficient general interest?

Good

Is the overall quality of the paper suitable?

Good

Can the paper be shortened without overall detriment to the main message?

Yes

Do you think some of the material would be more appropriate as an electronic appendix?

No

Do you have any ethical concerns with this paper?

No

Recommendation?

Accept as is

Comments to the Author(s)

The authors have edited all my comments. I have no further comments on the manuscript.

Review form: Referee 2

Is the manuscript an original and important contribution to its field?

Good

Is the paper of sufficient general interest?

Good

Is the overall quality of the paper suitable?

Good

Can the paper be shortened without overall detriment to the main message?

Yes

Do you think some of the material would be more appropriate as an electronic appendix?

No

Do you have any ethical concerns with this paper?

No

Recommendation?

Major revision is needed (please make suggestions in comments)

Comments to the Author(s)

The authors have done a good job addressing my original comments. I have some additional, mostly minor, comments that I believe the authors should address before the manuscript is ready for publication.

1. Pg. 5, ln. 39: "yields a Monte Carlo sample estimate of $Q(f)$ " My understanding is that you use a single set K to determine the quality of fit in this work. But the phrase "Monte Carlo sample" makes it seem like you are using multiple realizations of the set K to estimate the quality of fit statistics (hence Monte Carlo). If in fact multiple realizations of K are used to determine the metrics, can you clarify this in the text?

2. Eq. 3.1: As I understand it, Beta on the left hand side is a vector of length 2, for the horizontal and vertical wind components. Since it is a vector, should it be written in bold or with some other indication that it is a vector? Additionally, I believe "N" at the end of the equation should be "K".

3. Eq. 3.12: Why does "eta" not appear in this upper bound, despite it being part of the loss function? Further, a reference to the proof in the appendix is missing when presenting this inequality.

4. Pg. 8, ln. 55: "keeping tau, lambda, r, and eta fixed." What are the initial conditions for lambda and eta? The original values for the other variables are specified, but seem to be missing for these two.

5. Pg. 11, ln. 58: "the 5th percentile". Here and elsewhere (e.g., Table 1) use 5th percentile, but section 2c uses "95 percent". Should these all be "95th percentile"?

6. Fig. 6: This is a very informative plot. I would encourage you to add a similar (sub)plot showing the fraction of variance unexplained as well, since this would help show the relative performance of the interpolation models regardless of the variance of the wind for a particular season/time of day (and help support some of the discussion in Section 5).

7. Pg. 14, ln. 47: " $\text{Var}(\tilde{E}) = 0.1$ or a 1% relative error in $Q...$ ": The 95% confidence interval is given by 2 standard deviations, so this would yield a 95% confidence interval of +/- 2% relative error. But Section 2C explains that the requirement is to obtain a 95% confidence interval within 1% of the mean square wind speed. Why is there a discrepancy here? Further, Section 2c also says that a requirement is for the 95% confidence interval of the "difference" in variance unexplained to be within 1% of the mean square wind speed as well. Is this requirement met by the choice of $|K| = 500$?

8. Figs. 9 and 10: Consider adding a colorbar to these plots. Further, Fig. 9 says that the distribution is estimated from 1000 steps of the algorithm. Does this mean the full algorithm was run 1000 times (with $B = 500$), or was it run once using $B = 1000$?

9. Pg. 16, ln. 39: "likely improve the accuracy of this approximation" Can you clarify what approximation you are referring to here?

10. Pg. 17, ln. 7: "we derived an optimal density for the Fourier frequencies" As I understand it, you derived a density that minimizes an upper bound to the loss function, but it is not necessarily "optimal". Should this be rephrased?

11. Eq. 7.3: This loss function doesn't match the loss function used in the main part of the manuscript (Eq. 3.4). Please explain how this proof addresses the specific loss function (with Sobolev norm) in Eq. 3.4.

12. Pg. 18, ln. 46: "is exactly the variance of beta" Isn't this actually the variance of the error between beta and $f(x)$?

13. Pg. 18, ln. ~50: In the left hand side of the equation, should the expectation be conditioned on x , similar to Eq. 7.11?

14. Appendix: I understand the point of condensing the proof to keep the appendix concise, but there are several areas that I do not follow. I hope you can provide some clarification, even if it does not make it into the manuscript. First, in the equation in pg. 18, ln. ~50, I do not see where the " $1/K$ " in front of $E[|f(x)|^2]$ comes from.

15. Eq. 7.12: Can you explain how you get from the left hand side to the middle of this inequality?

16. Pg. 19, ln. 19: Can you define " r_m "? Without this, it is hard to follow the equation around pg. 19, ln. ~25, particularly the last equality.

17. Eq. 7.17: I can't find how the derivative of Eq. 7.16 simplifies to this form, since I end up with a summation term with $p(\omega)^3$ in the denominator, in addition to the one with $p(\omega)^4$ that is shown.

Minor/formatting comments:

1. Pg. 3, ln. 23: "pressure are used pose" -> "used to pose"

2. Fig. 1 caption: The degree symbol does not show up properly. Similar formatting issues occur throughout the manuscript (e.g., references 8, 19, 27)

3. Pg. 5, ln. 6: Here the phrase "hyper parameter" is used, but sometimes it is spelled as one word "hyperparameter". Please be consistent.

4. Eq. 3.9: I believe " x " on the left hand side should be bold.

5. Pg. 7, ln. 51: "the random Fourier..." capitalize "the"

6. Pg. 8, ln. 44: "north-west length" -> "east-west" or "north-south"?

7. Algorithm 1: The symbol " r ", the rounded elements of σ_{r_N} , is already used to describe a regularization parameter. Can a different symbol be used here?

8. Eq. 3.15: The equation uses $x - x$ prime, but the text above uses x prime - x . Just a suggestion to make them consistent.

9. Pg. 11, ln. 58: "two standard deviations of correspond": two standard deviations of what?

10. Fig. 5 caption: Capitalize "sweden". Further, the fractions of variance unexplained listed are 0.57, 0.55, 0.7. These don't match the values in Table 1. Are they just the metrics for this particular time step?

11. Pg. 18, ln. 6: "a family a family" typo

12. The multi line equations in the appendix are missing equation numbers (pg. 18, ln. ~50 and pg. 19, ln. ~25).

Pg. 20, ln. 24: Commas are missing between the author names

Decision letter (RSPA-2021-0236.R1)

11-Oct-2021

Dear Dr Ström,

On behalf of the Editor, I am pleased to inform you that your Manuscript RSPA-2021-0236.R1 entitled "Wind Field Reconstruction with Adaptive Random Fourier Features" has been accepted for publication subject to minor revisions in Proceedings A. Please find the referees' comments below.

The reviewer(s) have recommended publication, but also suggest some minor revisions to your manuscript. Therefore, I invite you to respond to the reviewer(s)' comments and revise your manuscript. Please note that we have a strict upper limit of 28 pages for each paper. Please endeavour to incorporate any revisions while keeping the paper within journal limits. Please note that page charges are made on all papers longer than 20 pages. If you cannot pay these charges you must reduce your paper to 20 pages before submitting your revision. Your paper has been ESTIMATED to be 21 pages. We cannot proceed with typesetting your paper without your agreement to meet page charges in full should the paper exceed 20 pages when typeset. If you have any questions, please do get in touch.

It is a condition of publication that you submit the revised version of your manuscript within 7 days. If you do not think you will be able to meet this date please let me know in advance of the due date.

To revise your manuscript, log into <https://mc.manuscriptcentral.com/prsa> and enter your Author Centre, where you will find your manuscript title listed under "Manuscripts with Decisions." Under "Actions," click on "Create a Revision." Your manuscript number has been appended to denote a revision.

You will be unable to make your revisions on the originally submitted version of the manuscript. Instead, revise your manuscript and upload a new version through your Author Centre.

When submitting your revised manuscript, you will be able to respond to the comments made by the referee(s) and upload a file "Response to Referees" in Step 1: "View and Respond to Decision Letter". Please provide a point-by-point response to the comments raised by the reviewers and the editor(s). A thorough response to these points will help us to assess your revision quickly. You can also upload a 'tracked changes' version either as part of the 'Response to reviews' or as a 'Main document'.

IMPORTANT: Your original files are available to you when you upload your revised manuscript. Please delete any redundant files before completing the submission process.

When uploading your revised files, please make sure that you include the following as we cannot proceed without these:

- 1) A text file of the manuscript (doc, txt, rtf or tex), including the references, tables (including captions) and figure captions. Please remove any tracked changes from the text before submission. PDF files are not an accepted format for the "Main Document".
- 2) A separate electronic file of each figure (tif, eps or print-quality pdf preferred). The format should be produced directly from original creation package, or original software format.
- 3) Electronic Supplementary Material (ESM): all supplementary materials accompanying an accepted article will be treated as in their final form. Note that the Royal Society will not edit or

typeset supplementary material and it will be hosted as provided. Please ensure that the supplementary material includes the paper details where possible (authors, article title, journal name). Supplementary files will be published alongside the paper on the journal website and posted on the online figshare repository (<https://figshare.com>). The heading and legend provided for each supplementary file during the submission process will be used to create the figshare page, so please ensure these are accurate and informative so that your files can be found in searches. Files on figshare will be made available approximately one week before the accompanying article so that the supplementary material can be attributed a unique DOI.

Alternatively you may upload a zip folder containing all source files for your manuscript as described above with a PDF as your "Main Document". This should be the full paper as it appears when compiled from the individual files supplied in the zip folder.

Article Funder

Please ensure you fill in the Article Funder question on page 2 to ensure the correct data is collected for FundRef (<http://www.crossref.org/fundref/>).

Media summary

Please ensure you include a short non-technical summary (up to 100 words) of the key findings/importance of your paper. This will be used for to promote your work and marketing purposes (e.g. press releases). The summary should be prepared using the following guidelines:

*Write simple English: this is intended for the general public. Please explain any essential technical terms in a short and simple manner.

*Describe (a) the study (b) its key findings and (c) its implications.

*State why this work is newsworthy, be concise and do not overstate (true 'breakthroughs' are a rarity).

*Ensure that you include valid contact details for the lead author (institutional address, email address, telephone number).

Cover images

We welcome submissions of images for possible use on the cover of Proceedings A. Images should be square in dimension and please ensure that you obtain all relevant copyright permissions before submitting the image to us. If you would like to submit an image for consideration please send your image to proceedingsa@royalsociety.org

Open Access

You are invited to opt for open access, our author pays publishing model. Payment of open access fees will enable your article to be made freely available via the Royal Society website as soon as it is ready for publication. For more information about open access please visit <https://royalsociety.org/journals/authors/open-access/>. The open access fee for this journal is £1700/\$2380/€2040 per article. VAT will be charged where applicable. Please note that if the corresponding author is at an institution that is part of a Read and Publishing deal you are required to select this option. See <https://royalsociety.org/journals/librarians/purchasing/read-and-publish/read-publish-agreements/> for further details.

Once again, thank you for submitting your manuscript to Proceedings A and I look forward to receiving your revision. If you have any questions at all, please do not hesitate to get in touch.

Best wishes
Raminder Shergill
proceedingsa@royalsociety.org

Proceedings A

on behalf of
Professor Johannes Zimmer
Board Member
Proceedings A

Reviewer(s)' Comments to Author:

Referee: 1

Comments to the Author(s)

The authors have edited all my comments. I have no further comments on the manuscript.

Referee: 2

Comments to the Author(s)

The authors have done a good job addressing my original comments. I have some additional, mostly minor, comments that I believe the authors should address before the manuscript is ready for publication.

1. Pg. 5, ln. 39: "yields a Monte Carlo sample estimate of $Q(f)$ " My understanding is that you use a single set K to determine the quality of fit in this work. But the phrase "Monte Carlo sample" makes it seem like you are using multiple realizations of the set K to estimate the quality of fit statistics (hence Monte Carlo). If in fact multiple realizations of K are used to determine the metrics, can you clarify this in the text?

2. Eq. 3.1: As I understand it, β on the left hand side is a vector of length 2, for the horizontal and vertical wind components. Since it is a vector, should it be written in bold or with some other indication that it is a vector? Additionally, I believe " N " at the end of the equation should be " K ".

3. Eq. 3.12: Why does " η " not appear in this upper bound, despite it being part of the loss function? Further, a reference to the proof in the appendix is missing when presenting this inequality.

4. Pg. 8, ln. 55: "keeping τ , λ , r , and η fixed." What are the initial conditions for λ and η ? The original values for the other variables are specified, but seem to be missing for these two.

5. Pg. 11, ln. 58: "the 5th percentile". Here and elsewhere (e.g., Table 1) use 5th percentile, but section 2c uses "95 percent". Should these all be "95th percentile"?

6. Fig. 6: This is a very informative plot. I would encourage you to add a similar (sub)plot showing the fraction of variance unexplained as well, since this would help show the relative performance of the interpolation models regardless of the variance of the wind for a particular season/time of day (and help support some of the discussion in Section 5).

7. Pg. 14, ln. 47: " $\text{Var}(\tilde{E}) = 0.1$ or a 1% relative error in $Q\dots$ ": The 95% confidence interval is given by 2 standard deviations, so this would yield a 95% confidence interval of $\pm 2\%$ relative error. But Section 2C explains that the requirement is to obtain a 95% confidence interval within 1% of the mean square wind speed. Why is there a discrepancy here? Further, Section 2c also says that a requirement is for the 95% confidence interval of the "difference" in variance unexplained to be within 1% of the mean square wind speed as well. Is this requirement met by the choice of $|K| = 500$?

8. Figs. 9 and 10: Consider adding a colorbar to these plots. Further, Fig. 9 says that the distribution is estimated from 1000 steps of the algorithm. Does this mean the full algorithm was run 1000 times (with $B = 500$), or was it run once using $B = 1000$?

9. Pg. 16, ln. 39: "likely improve the accuracy of this approximation" Can you clarify what approximation you are referring to here?

10. Pg. 17, ln. 7: "we derived an optimal density for the Fourier frequencies" As I understand it, you derived a density that minimizes an upper bound to the loss function, but it is not necessarily "optimal". Should this be rephrased?

11. Eq. 7.3: This loss function doesn't match the loss function used in the main part of the manuscript (Eq. 3.4). Please explain how this proof addresses the specific loss function (with Sobolev norm) in Eq. 3.4.

12. Pg. 18, ln. 46: "is exactly the variance of beta" Isn't this actually the variance of the error between beta and $f(x)$?

13. Pg. 18, ln. ~50: In the left hand side of the equation, should the expectation be conditioned on x , similar to Eq. 7.11?

14. Appendix: I understand the point of condensing the proof to keep the appendix concise, but there are several areas that I do not follow. I hope you can provide some clarification, even if it does not make it into the manuscript. First, in the equation in pg. 18, ln. ~50, I do not see where the " $1/K$ " in front of $E[|f(x)|^2]$ comes from.

15. Eq. 7.12: Can you explain how you get from the left hand side to the middle of this inequality?

16. Pg. 19, ln. 19: Can you define " r_m "? Without this, it is hard to follow the equation around pg. 19, ln. ~25, particularly the last equality.

17. Eq. 7.17: I can't find how the derivative of Eq. 7.16 simplifies to this form, since I end up with a summation term with $p(\omega)^3$ in the denominator, in addition to the one with $p(\omega)^4$ that is shown.

Minor/formatting comments:

1. Pg. 3, ln. 23: "pressure are used pose" -> "used to pose"

2. Fig. 1 caption: The degree symbol does not show up properly. Similar formatting issues occur throughout the manuscript (e.g., references 8, 19, 27)

3. Pg. 5, ln. 6: Here the phrase "hyper parameter" is used, but sometimes it is spelled as one word "hyperparameter". Please be consistent.

4. Eq. 3.9: I believe " x " on the left hand side should be bold.

5. Pg. 7, ln. 51: "the random Fourier..." capitalize "the"

6. Pg. 8, ln. 44: "north-west length" -> "east-west" or "north-south"?

7. Algorithm 1: The symbol " r ", the rounded elements of σ_{r_N} , is already used to describe a regularization parameter. Can a different symbol be used here?

8. Eq. 3.15: The equation uses $x - x$ prime, but the text above uses x prime - x . Just a suggestion to make them consistent.

9. Pg. 11, ln. 58: "two standard deviations of correspond": two standard deviations of what?

10. Fig. 5 caption: Capitalize "sweden". Further, the fractions of variance unexplained listed are 0.57, 0.55, 0.7. These don't match the values in Table 1. Are they just the metrics for this particular time step?

11. Pg. 18, ln. 6: "a family a family" typo

12. The multi line equations in the appendix are missing equation numbers (pg. 18, ln. ~50 and pg. 19, ln. ~25).

Pg. 20, ln. 24: Commas are missing between the author names

Board Member

Comments to Author(s):

The second reviewer has a substantial list of suggested changes, and their implementation will improve the manuscript. Please address these suggestions wherever possible.

Author's Response to Decision Letter for (RSPA-2021-0236.R1)

See Appendix B.

Decision letter (RSPA-2021-0236.R2)

20-Oct-2021

Dear Dr Ström

I am pleased to inform you that your manuscript entitled "Wind Field Reconstruction with Adaptive Random Fourier Features" has been accepted in its final form for publication in Proceedings A.

Our Production Office will be in contact with you in due course. You can expect to receive a proof of your article soon. Please contact the office to let us know if you are likely to be away from e-mail in the near future. If you do not notify us and comments are not received within 5 days of sending the proof, we may publish the paper as it stands.

As a reminder, you have provided the following 'Data accessibility statement' (if applicable). Please remember to make any data sets live prior to publication, and update any links as needed when you receive a proof to check. It is good practice to also add data sets to your reference list. *Statement (if applicable)*: Python code for this paper can be found on Bitbucket via the following url: https://bitbucket.org/the_eye_of_the_breeze/div-fourier-publication/src/master/. It implements interpolation methods discussed in section 3 and contains the data set used for training and evaluation.

Open access

You are invited to opt for open access, our author pays publishing model. Payment of open access fees will enable your article to be made freely available via the Royal Society website as soon as it is ready for publication. For more information about open access please visit

<https://royalsociety.org/journals/authors/which-journal/open-access/>. The open access fee for this journal is £1700/\$2380/€2040 per article. VAT will be charged where applicable.

Note that if you have opted for open access then payment will be required before the article is published – payment instructions will follow shortly.

If you wish to opt for open access then please inform the editorial office (proceedingsa@royalsociety.org) as soon as possible.

Your article has been estimated as being 21 pages long. Our Production Office will inform you of the exact length at the proof stage.

Proceedings A levies charges for articles which exceed 20 printed pages. (based upon approximately 540 words or 2 figures per page). Articles exceeding this limit will incur page charges of £150 per page or part page, plus VAT (where applicable).

Under the terms of our licence to publish you may post the author generated postprint (ie. your accepted version not the final typeset version) of your manuscript at any time and this can be made freely available. Postprints can be deposited on a personal or institutional website, or a recognised server/repository. Please note however, that the reporting of postprints is subject to a media embargo, and that the status the manuscript should be made clear. Upon publication of the definitive version on the publisher's site, full details and a link should be added.

You can cite the article in advance of publication using its DOI. The DOI will take the form: 10.1098/rspa.XXXX.YYYY, where XXXX and YYYY are the last 8 digits of your manuscript number (eg. if your manuscript number is RSPA-2017-1234 the DOI would be 10.1098/rspa.2017.1234).

For tips on promoting your accepted paper see our blog post:
<https://royalsociety.org/blog/2020/07/promoting-your-latest-paper-and-tracking-your-results/>

On behalf of the Editor of Proceedings A, we look forward to your continued contributions to the Journal.

Sincerely,
Raminder Shergill
proceedingsa@royalsociety.org

on behalf of
Professor Johannes Zimmer
Board Member
Proceedings A

Appendix A

Responses to Comments by Referees.

Jonas Kiessling

Emanuel Ström

Raul Tempone

October 18, 2021

Abstract

This document is a point-by point response to the comments by the referees. We would like to take the opportunity to thank the referees for valuable feedback. The references to equations and figures in this document are colored **green** when they refer to the numbering in the second revision of the manuscript, and **blue** when they refer to the numbering in the first revision of the manuscript. References to literature and references that are the same for both manuscripts are black. We have also attached a document showing the changes made to the manuscript, in which removed text is marked in **blue**, and added text is marked in **green**. Lastly,

whenever changes to the manuscript are quoted, they are put in a boxed environment for clarity.

Referee 1

Comment 1 (Summary) *Comments to the Author(s) The authors have edited all my comments. I have no further comments on the manuscript.*

Referee 2

Comment 2 (Summary) *The authors have done a good job addressing my original comments. I have some additional, mostly minor, comments that I believe the authors should address before the manuscript is ready for publication.*

We thank the referee for the valuable comments, it is clear to us that a lot of care was put into reading our manuscript and we are very thankful for this.

Comment 3 (Pg.5, ln. 39) *"yields a Monte Carlo sample estimate of $Q(f)$ " My understanding is that you use a single set K to determine the quality of fit in this work. But the phrase "Monte Carlo sample" makes it seem like you are using multiple realizations of the set K to estimate the quality of fit statistics (hence Monte Carlo). If in fact multiple realizations of K are used to determine the metrics, can you clarify this in the text?*

The set K is the set of Monte Carlo samples. It is a random subset of the data.

Comment 4 (Eq. 3.1) *As I understand it, Beta on the left hand side is a vector of length 2, for the horizontal and vertical wind components. Since it is a vector, should it be written in bold or with some other indication that it is a vector? Additionally, I believe "N" at the end of the equation should be "K".*

The "N" should indeed be a "K". We also clarified in Eq 3.1 that we extract only the real part of the Fourier series, not the complex part. We have done our best to keep clear and consistent notation throughout the manuscript. To further clarify the difference between $\boldsymbol{\beta}$ and β and between $\boldsymbol{\omega}$ and ω , we have added a sentence below (3.1):

With slight abuse of notation, we let β_k and ω_k denote two-dimensional vectors in C^2 and R^2 respectively. Furthermore, $\boldsymbol{\omega} = (\omega_1, \omega_2, \dots, \omega_K)$ and $\boldsymbol{\beta} = (\beta_1, \beta_2, \dots, \beta_K)$.

Comment 5 (Eq. 3.12) *Why does "eta" not appear in this upper bound, despite it being part of the loss function? Further, a reference to the proof in the appendix is missing when presenting this inequality.*

Added a reference to the proof in the appendix. We adjusted the inequality to include η . The argument for including multiple regularisation parameters is analogous to the proof when only using one regularisation parameter, and we included a short comment before the proof to clarify this:

By replacing $\lambda\bar{C}$ in (7.7) with $\sum_{s=1}^S \lambda_s \bar{C}_s$, Proposition 1 can be generalised to include a linear combination of S regularisation functions $\sum_{s=1}^S \lambda_s \|\mathcal{L}_s \beta\|^2$.

Comment 6 (Pg. 8, ln. 55) *"keeping tau, lambda, r, and eta fixed." What are the initial conditions for lambda and eta? The original values for the other variables are specified, but seem to be missing for these two.*

This is a good point. Referring to our code, we found a small mistake in this description. We corrected this mistake in the manuscript:

The regularisation parameters λ and η were obtained by running a grid search to minimise the variance unexplained in (2.4), while keeping τ, r, γ and σ fixed as specified above. Since the values for γ and σ are only optimal in the limit, a second grid search was done in a neighbourhood of the limit values for γ and σ , while keeping τ, r, λ and η fixed.

Comment 7 (Pg. 11, ln. 58) *"the 5th percentile". Here and elsewhere (e.g., Table 1) use 5th percentile, but section 2c uses "95 percent". Should these all be "95th percentile"?*

The expression "5th percentile confidence interval" was mistakenly used to mean "95 percent confidence interval". We have now replaced the former with the latter everywhere in the manuscript.

Comment 8 (Fig. 6) *This is a very informative plot. I would encourage you to add a similar (sub)plot showing the fraction of variance unexplained as well, since this would help show the relative performance of the interpolation models regardless of the variance of the wind for a particular season/time of day (and help support some of the discussion in Section 5).*

We agree that a figure showing relative errors would be an informative addition. In the interest of limiting page count however, we choose not to include the figure. We still feel like Figure 6 is sufficient to clearly show the differences in behaviour between the winter and summer months. The relative error is easily deduced from Figure 6, since the zero model error is included in the plot.

Comment 9 (Pg. 14, ln. 47) *"Var(tilde E) = 0.1 or a 1% relative error in Q...": The 95% confidence interval is given by 2 standard deviations, so this would yield a 95% confidence interval of +/- 2% relative error. But Section 2C explains that the requirement is to obtain a 95% confidence interval within 1% of the mean square wind speed. Why is there a discrepancy here? Further, Section 2c also says that a requirement is for the 95% confidence interval of the "difference" in variance unexplained to be within 1% of the mean square wind speed as well. Is this requirement met by the choice of $|K| = 500$?*

This is a good point. $\text{Var}(\tilde{E}) = 0.1$ was a typo. For increased clarity we replaced this sentence with a reference to the results table. It now reads:

Tables 1 and 2, show that this sample size achieves the goal of 1% relative error with 95% confidence for both Q and ΔQ that we formulated in Section 2 (c). The only exception is the Nearest Neighbors model, for which the confidence interval for the error estimate is 2%. This is still sufficient accuracy for our purpose.

Comment 10 (Figs. 9 and 10) *Consider adding a colorbar to these plots. Further, Fig. 9 says that the distribution is estimated from 1000 steps of the algorithm. Does this mean the full algorithm was run 1000 times (with $B = 500$), or was it run once using $B = 1000$?*

We feel that a colorbar is not appropriate in this case, since Figure 5 and 6 have different ranges. With regards to your second point, we changed the figure description to avoid ambiguity. It now reads:

Algorithm 1 was run with $B = 1000$, collecting the Fourier frequencies at each step into a histogram.

Comment 11 (Pg. 16, ln. 39) *"likely improve the accuracy of this approximation" Can you clarify what approximation you are referring to here?*

Changed the sentence to:

If we were to transition from a 2D to a 3D setting by incorporating elevation as a feature in the random Fourier features, divergence penalisation would likely be more useful.

Comment 12 (Pg. 17, ln. 7) *"we derived an optimal density for the Fourier frequencies" As I understand it, you derived a density that minimizes an upper bound to the loss function, but it is not necessarily "optimal". Should this be rephrased?*

We rephrased this sentence as follows:

we derived an upper bound for the mean square error of the random Fourier Frequencies model, found a density that minimised this bound and devised an adaptive Metropolis algorithm for sampling from this density.

Comment 13 (Eq. 7.3) *This loss function doesn't match the loss function used in the main part of the manuscript (Eq. 3.4). Please explain how this proof addresses the specific loss function (with Sobolev norm) in Eq. 3.4.*

We include a sentence before the proof on how to generalise the proof to sums of regularisers:

By replacing $\lambda\bar{C}$ in (7.7) with $\sum_{s=1}^S \lambda_s \bar{C}_s$, Proposition 1 can be generalised to include a linear combination of S regularisation functions $\sum_{s=1}^S \lambda_s \|\mathcal{L}_s \beta\|^2$.

Furthermore, we add a definition (1) of the Sobolev norm that clarifies its connection to the one formulated in Proposition 1:

The expression $\|\beta\|_{S(2,2)}^2$ denotes the second order squared Sobolev-norm of β [1]:

$$\|\beta\|_{S(2,2)}^2 = \|\beta\|^2 + r (\|\partial_x \beta\|^2 + \|\partial_y \beta\|^2) + r^2 (\|\partial_{xx} \beta\|^2 + 2\|\partial_x \partial_y \beta\|^2 + \|\partial_{yy} \beta\|^2) \quad (1)$$

Comment 14 (Pg. 18, ln. 46) *"is exactly the variance of beta" Isn't this actually the variance of the error between beta and f(x)?*

It is both, since beta is unbiased by construction in this part of the proof. To clarify what we mean by the variance of β , we made some slight changes to this part of the proof:

β is then a function of the random variables \mathbf{x} and $\boldsymbol{\omega}$, which we denote $\beta(\mathbf{x}; \boldsymbol{\omega})$. With this definition, β is unbiased given \mathbf{x} :

$$E \left[\beta(\mathbf{x}; \boldsymbol{\omega}) \mid \mathbf{x} \right] = \sum_{\boldsymbol{\omega} \in Z^2} K \frac{\hat{f}(\boldsymbol{\omega})}{2\pi K \rho(\boldsymbol{\omega})} e^{i\boldsymbol{\omega} \cdot \mathbf{x}} \rho(\boldsymbol{\omega}) = \frac{1}{2\pi} \sum_{\boldsymbol{\omega}' \in Z^2} \hat{f}(\boldsymbol{\omega}') e^{i\boldsymbol{\omega}' \cdot \mathbf{x}} = f(\mathbf{x}), \quad (2)$$

where we used that the components of $\boldsymbol{\omega}$ are iid. Using independence of the residuals ϵ and the data \mathbf{x} , we see that:

$$E \left[\|\beta(\mathbf{x}; \boldsymbol{\omega}) - \mathbf{u}\|^2 \mid \mathbf{x} \right] = E \left[\|\beta(\mathbf{x}; \boldsymbol{\omega}) - (f(\mathbf{x}) + \epsilon)\|^2 \mid \mathbf{x} \right] = E \left[\|\beta(\mathbf{x}; \boldsymbol{\omega}) - f(\mathbf{x})\|^2 \mid \mathbf{x} \right] + \sigma^2, \quad (3)$$

showing that the expected square error of β as a function of \mathbf{x} is the variance of β with respect to the frequencies $\boldsymbol{\omega}$, plus the variance of the noise."

Comment 15 (Pg. 18, ln. 50) : In the left hand side of the equation, should the expectation be conditioned on x , similar to Eq. 7.11?

In this case, no. The conditional expectation is an important step in simplifying the left hand side however, by using the tower property $E[Y] = E[E[Y|X]]$. This step has been left out to shorten the proof.

Comment 16 (Appendix) I understand the point of condensing the proof to keep the appendix concise, but there are several areas that I do not follow. I hope you can provide some clarification, even if it does not make it into the manuscript. First, in the equation in pg. 18, ln. 50, I do not see where the "1/K" in front of $E[|f(x)|^2]$ comes from.

We apologise for the brevity and as you say, the point is to keep the appendix as short as possible. Here a longer version of the derivation, hopefully it will clear up some of your questions:

$$\begin{aligned} E \left[\left\| \sum_{k=1}^K \beta_k e^{i\omega_k \cdot \mathbf{x}} - f(\mathbf{x}) \right\|^2 \right] & (1) = E \left[\left\| \sum_{k=1}^K \left(\beta_k e^{i\omega_k \cdot \mathbf{x}} - \frac{f(\mathbf{x})}{K} \right) \right\|^2 \right] \\ & (2) = \frac{1}{K^2} E \left[\left\| \sum_{k=1}^K (K \beta_k e^{i\omega_k \cdot \mathbf{x}} - f(\mathbf{x})) \right\|^2 \right] \\ & (3) = \frac{1}{K} E \left[\left\| K \beta_0 e^{i\omega_0 \cdot \mathbf{x}} - f(\mathbf{x}) \right\|^2 \right] \\ & (4) = K E \left[\|\beta_0\|^2 \right] - \frac{1}{K} E \left[\|f(\mathbf{x})\|^2 \right] \\ & (5) = \frac{1}{(2\pi)^2 K} E \left[\frac{\|\hat{f}(\omega_0)\|^2}{\rho(\omega_0)^2} \right] - \frac{1}{K} E \left[\|f(\mathbf{x})\|^2 \right] \\ & (6) \leq \frac{1}{(2\pi)^2 K} \sqrt{E \left[\frac{\|\hat{f}(\omega_0)\|^4}{\rho(\omega_0)^4} \right]} - \frac{1}{K} E \left[\|f(\mathbf{x})\|^2 \right]. \end{aligned}$$

In step (1) we move f inside the sum. In step (2), we factor out $1/K^2$. In step (3), we use the independence of the Fourier frequencies to move the sum outside the expectation without getting a "double sum". In step (4), we have used the property that $E[K\beta e^{i\omega \cdot \mathbf{x}}] = f(\mathbf{x})$ to separate the two terms inside the norm, and then factored out the K^2 from the β -term. In step (5), we use the definition of β_0 , to simplify the left term. In step (6) we use Jenessen's inequality to approximate the left term so that we get an expectation that matches with the term from the regulariser later on.

Comment 17 (Eq. 7.12) *Can you explain how you get from the left hand side to the middle of this inequality?*

Thanks to your question, we noticed a slight mistake in the derivation. We amended this section of the proof, which now reads:

Applying \mathcal{L} to $\beta(\mathbf{x})$ is equivalent to multiplying each term $\widehat{\beta}_{kj} e^{i\omega_k x}$, $j = 1, 2$ of the Fourier series with $\ell_j(\omega_k) = \sum_{m=1}^M c_m (i\omega_{k,1})^{\alpha_{m,1}} (i\omega_{k,2})^{\alpha_{m,2}}$, a multivariate polynomial in ω_k of degree d . Define $r(\omega) = |\ell_1(\omega)|^2 + |\ell_2(\omega)|^2$. Note that r has degree $2d$. It follows that

$$E [|\mathcal{L}\beta|^2] = \int_X E [|\mathcal{L}\beta(\mathbf{x}; \boldsymbol{\omega})|^2 | \mathbf{x}] d\mathbf{x} = K(2\pi)^2 E [|\ell_1(\omega_k) \widehat{\beta}_{k1} + \ell_2(\omega_k) \widehat{\beta}_{k2}|^2]. \quad (4)$$

The first equality comes from switching the order of integration, and in the second we use independence of the Fourier frequencies combined with the size $(2\pi)^2$ of the region $X = [0, 2\pi] \times [0, 2\pi]$. Applying the Cauchy-Schwartz inequality twice on the last expression of (4) results in:

$$K(2\pi)^2 E [|\ell_1(\omega_k) \beta_{k1} + \ell_2(\omega_k) \beta_{k2}|^2] \leq K E [r(\omega) |2\pi\beta|^2] \leq \frac{1}{K} \sqrt{E [r(\omega)^2] E \left[\frac{\|\hat{f}(\omega)\|^4}{\rho(\omega)^4} \right]} \quad (5)$$

Here is a line-by-line explanation of equation 7.12.

$$\begin{aligned}
E [|\mathcal{L}\beta|^2] & \quad (1) = E \left[\int_X |\mathcal{L}\beta(x)|^2 dx \right] \\
& \quad (2) = \int_X E [|\mathcal{L}\beta(x)|^2 | x] dx \\
& \quad (3) = \int_X E \left[\left| \sum_{k=1}^K (\ell_1(\omega_k)\beta_{k1} + \ell_2(\omega_k)\beta_{k2}) e^{i\omega_k \cdot x} \right|^2 \right] dx \\
& \quad (4) = \int_X E \left[K |(\ell_1(\omega_0)\beta_{01} + \ell_2(\omega_0)\beta_{02}) e^{i\omega_0 \cdot x}|^2 \right] dx \\
& \quad (5) = \int_X E \left[K |(\ell_1(\omega_0)\beta_{01} + \ell_2(\omega_0)\beta_{02})|^2 \right] dx \\
& \quad (6) = (2\pi)^2 K E \left[|(\ell_1(\omega_0)\beta_{01} + \ell_2(\omega_0)\beta_{02})|^2 \right] \\
& \quad (7) \leq K E [r(\omega_0) |2\pi\beta_{02}|^2].
\end{aligned}$$

Step (1) is the definition of the norm $|\mathcal{L}\beta|$. In step (2), we switch order of integration. In step (3), we expand the expression $\mathcal{L}\beta$ as a Fourier series. In step (4), we use independence of the Fourier frequencies. In step (5), we use that $|e^{i\omega \cdot x}|^2 = 1$. In (6), we use that the integrand is independent of x . Lastly, we use Cauchy-Schwarz. We have amended the proof accordingly.

Comment 18 (Pg. 19, ln. 19) *Can you define "r_m"? Without this, it is hard to follow the equation around pg. 19, ln. 25, particularly the last equality.*

We define r_m in the first revision of the manuscript, below equation 7.13 as the coefficients of the polynomial $r(\omega)^2$:

The multivariate polynomial $r(\omega)^2 := \sum_{m=1}^{M'} r_m \omega_1^{\alpha_{m,1}} \omega_2^{\alpha_{m,2}}$ is of order $4d$, i.e. $\alpha_{m,1} + \alpha_{m,2} \leq 4d$.

$r(\omega)$ was in turn defined as $|\ell_1(\omega)|^2 + |\ell_2(\omega)|^2$ above equation 7.12. We added some clarification to the steps.

Comment 19 (Eq. 7.17) *I can't find how the derivative of Eq. 7.16 simplifies to this form, since I end up with a summation term with $p(\omega)^3$ in the denominator, in addition to the one with $p(\omega)^4$ that is shown.*

The term with $p(\omega)^3$ is denoted as "c₂" in our proof. Here is a more detailed derivation that shows what happens to this term.

$$H'(\epsilon) = -3 \left(\sum_{\omega \in Z^2} \frac{\|\hat{f}(\omega)\|^4}{(p(\omega) + \epsilon \delta(\omega))^4} \delta(\omega) \right) \left(\sum_{\omega \in Z^2} p(\omega) \right)^3 + \\ + 3 \left(\sum_{\omega \in Z^2} \frac{\|\hat{f}(\omega)\|^4}{(p(\omega) + \epsilon \delta(\omega))^3} \right) \left(\sum_{\omega \in Z^2} p(\omega) \right)^2 \sum_{\omega \in Z^2} \delta(\omega)$$

Evaluating at $\epsilon = 0$:

$$H'(0) = -3 \left(\sum_{\omega \in Z^2} \frac{\|\hat{f}(\omega)\|^4}{p(\omega)^4} \delta(\omega) \right) \overbrace{\left(\sum_{\omega \in Z^2} p(\omega) \right)^3}^{:=c_1} + \\ + 3 \underbrace{\left(\sum_{\omega \in Z^2} \frac{\|\hat{f}(\omega)\|^4}{p(\omega)^3} \right)}_{:=c_2} \left(\sum_{\omega \in Z^2} p(\omega) \right)^2 \sum_{\omega \in Z^2} \delta(\omega)$$

We define the constants c_1 and c_2 as above. Then, setting $H'(0) = 0$ we get

$$\sum_{\omega \in Z^2} \left(\frac{\|\hat{f}(\omega)\|^4}{p(\omega)^4} - \frac{c_2}{c_1} \right) \delta(\omega) = 0$$

And since $\delta(\omega)$ is arbitrary, and p sums to 1, we must have

$$\frac{\|\hat{f}(\omega)\|^4}{p(\omega)^4} - \frac{c_2}{c_1} \iff p(\omega) \propto \|\hat{f}(\omega)\|. \iff p(\omega) = \frac{\|\hat{f}(\omega)\|}{\sum_{\omega \in Z^2} \|\hat{f}(\omega)\|}.$$

Minor/formatting

Comment 20 (Pg. 3, ln. 23) "pressure are used pose" -j "used to pose"

Changed to

assumptions about wind field, temperature and pressure are expressed as a set of differential equations that describe how the system evolves over time.

Comment 21 (Fig. 1 caption) The degree symbol does not show up properly. Similar formatting issues occur throughout the manuscript (e.g., references 8, 19, 27)

This is due to "special characters" like °,ö,ä,å not showing up properly when compiling the document. We suspect this has to do with the way that the document is compiled in ScholarOne, as it works fine in overleaf. We believe this issue is resolved now.

Comment 22 (Pg. 5, ln. 6) *Here the phrase "hyper parameter" is used, but sometimes it is spelled as one word "hyperparameter". Please be consistent.*

Changed to hyper parameter everywhere.

Comment 23 (Eq. 3.9) *I believe "x" on the left hand side should be bold.*

Changed left hand side "x" to bold

Comment 24 (Pg. 7, ln. 51) *"the random Fourier..." capitalize "the"*

Capitalised "the".

Comment 25 (Pg. 8, ln. 44) *"north-west length" -j "east-west" or "north-south"?*

changed to "north-south".

Comment 26 (Algorithm 1) *The symbol "r", the rounded elements of sigma r_N , is already used to describe a regularization parameter. Can a different symbol be used here?*

Changed to δ_N and δ .

Comment 27 (Eq. 3.15) *The equation uses $x - x'$, but the text above uses x prime - x . Just a suggestion to make them consistent.*

Switched the text above to $x - x'$.

Comment 28 (Pg. 11, ln. 58) *"two standard deviations of correspond": two standard deviations of what?*

Changed to:

means that two standard deviations of \mathcal{E} and \mathcal{Q} correspond to 95 % confidence intervals for \mathcal{E} and \mathcal{Q} , respectively.

Comment 29 (Fig. 5 caption) *Capitalize "sweden". Further, the fractions of variance unexplained listed are 0.57, 0.55, 0.7. These don't match the values in Table 1. Are they just the metrics for this particular time step?*

Capitalised Sweden. They are the fraction of variance unexplained for this particular time step. We clarified this:

From left to right, the fraction of variance unexplained for the above methods at this specific time is 0.57, 0.55 and 0.70.

Comment 30 (Pg. 18, ln. 6) *"a family a family" typo*

Fixed.

Comment 31 *The multi line equations in the appendix] are missing equation numbers (pg. 18, ln. 50 and pg. 19, ln. 25).*

Added equation numbers.

Comment 32 (Pg. 20, ln. 24) *Commas are missing between the author names.*

Additional changes

Apart from the above comments we made the following minor changes to the document:

1. we replaced the reference [2] with a published version, [3]
2. We corrected an error in the Funding statement. The funding statement now reads as follows.

Jonas Kiessling and Raúl Tempone were partially supported by the KAUST Office of Sponsored Research (OSR) under Award numbers OSR-2019-CRG8-4033 and OSR-2019-CRG8-4033.2, and the Alexander von Humboldt Foundation, through the Alexander von Humboldt Professorship award. Emanuel Ström was supported by the KAUST Visiting Student Research Program (VSRP).

References

- [1] Philippe Blanchard and Erwin Bruning. *Mathematical Methods in Physics*, volume 69 of *Progress in Mathematical Physics*. Birkhäuser, 2nd edition, 2015.
- [2] Luis Espath, Dmitry Kabanov, Jonas Kiessling, and Raúl Tempone. Statistical learning for fluid flows: Sparse fourier divergence-free approximations, 2021.
- [3] Luis Espath, Dmitry Kabanov, Jonas Kiessling, and Raúl Tempone. Statistical learning for fluid flows: Sparse fourier divergence-free approximations. *Physics of Fluids*, 33(9):097108, 2021.

Appendix B

Responses to Comments by Referees.

Jonas Kiessling

Emanuel Ström

Raul Tempone

October 18, 2021

Abstract

This document is a point-by point response to the comments by the referees. We would like to take the opportunity to thank the referees for valuable feedback. The references to equations and figures in this document are colored **green** when they refer to the numbering in the second revision of the manuscript, and **blue** when they refer to the numbering in the first revision of the manuscript. References to literature and references that are the same for both manuscripts are black. We have also attached a document showing the changes made to the manuscript, in which removed text is marked in **blue**, and added text is marked in **green**. Lastly,

whenever changes to the manuscript are quoted, they are put in a boxed environment for clarity.

Referee 1

Comment 1 (Summary) *Comments to the Author(s) The authors have edited all my comments. I have no further comments on the manuscript.*

Referee 2

Comment 2 (Summary) *The authors have done a good job addressing my original comments. I have some additional, mostly minor, comments that I believe the authors should address before the manuscript is ready for publication.*

We thank the referee for the valuable comments, it is clear to us that a lot of care was put into reading our manuscript and we are very thankful for this.

Comment 3 (Pg.5, ln. 39) *"yields a Monte Carlo sample estimate of $Q(f)$ " My understanding is that you use a single set K to determine the quality of fit in this work. But the phrase "Monte Carlo sample" makes it seem like you are using multiple realizations of the set K to estimate the quality of fit statistics (hence Monte Carlo). If in fact multiple realizations of K are used to determine the metrics, can you clarify this in the text?*

The set K is the set of Monte Carlo samples. It is a random subset of the data.

Comment 4 (Eq. 3.1) *As I understand it, Beta on the left hand side is a vector of length 2, for the horizontal and vertical wind components. Since it is a vector, should it be written in bold or with some other indication that it is a vector? Additionally, I believe "N" at the end of the equation should be "K".*

The "N" should indeed be a "K". We also clarified in Eq 3.1 that we extract only the real part of the Fourier series, not the complex part. We have done our best to keep clear and consistent notation throughout the manuscript. To further clarify the difference between $\boldsymbol{\beta}$ and β and between $\boldsymbol{\omega}$ and ω , we have added a sentence below (3.1):

With slight abuse of notation, we let β_k and ω_k denote two-dimensional vectors in C^2 and R^2 respectively. Furthermore, $\boldsymbol{\omega} = (\omega_1, \omega_2, \dots, \omega_K)$ and $\boldsymbol{\beta} = (\beta_1, \beta_2, \dots, \beta_K)$.

Comment 5 (Eq. 3.12) *Why does "eta" not appear in this upper bound, despite it being part of the loss function? Further, a reference to the proof in the appendix is missing when presenting this inequality.*

Added a reference to the proof in the appendix. We adjusted the inequality to include η . The argument for including multiple regularisation parameters is analogous to the proof when only using one regularisation parameter, and we included a short comment before the proof to clarify this:

By replacing $\lambda\bar{C}$ in (7.7) with $\sum_{s=1}^S \lambda_s \bar{C}_s$, Proposition 1 can be generalised to include a linear combination of S regularisation functions $\sum_{s=1}^S \lambda_s \|\mathcal{L}_s \beta\|^2$.

Comment 6 (Pg. 8, ln. 55) *"keeping tau, lambda, r, and eta fixed." What are the initial conditions for lambda and eta? The original values for the other variables are specified, but seem to be missing for these two.*

This is a good point. Referring to our code, we found a small mistake in this description. We corrected this mistake in the manuscript:

The regularisation parameters λ and η were obtained by running a grid search to minimise the variance unexplained in (2.4), while keeping τ, r, γ and σ fixed as specified above. Since the values for γ and σ are only optimal in the limit, a second grid search was done in a neighbourhood of the limit values for γ and σ , while keeping τ, r, λ and η fixed.

Comment 7 (Pg. 11, ln. 58) *"the 5th percentile". Here and elsewhere (e.g., Table 1) use 5th percentile, but section 2c uses "95 percent". Should these all be "95th percentile"?*

The expression "5th percentile confidence interval" was mistakenly used to mean "95 percent confidence interval". We have now replaced the former with the latter everywhere in the manuscript.

Comment 8 (Fig. 6) *This is a very informative plot. I would encourage you to add a similar (sub)plot showing the fraction of variance unexplained as well, since this would help show the relative performance of the interpolation models regardless of the variance of the wind for a particular season/time of day (and help support some of the discussion in Section 5).*

We agree that a figure showing relative errors would be an informative addition. In the interest of limiting page count however, we choose not to include the figure. We still feel like Figure 6 is sufficient to clearly show the differences in behaviour between the winter and summer months. The relative error is easily deduced from Figure 6, since the zero model error is included in the plot.

Comment 9 (Pg. 14, ln. 47) *"Var(tilde E) = 0.1 or a 1% relative error in Q...": The 95% confidence interval is given by 2 standard deviations, so this would yield a 95% confidence interval of +/- 2% relative error. But Section 2C explains that the requirement is to obtain a 95% confidence interval within 1% of the mean square wind speed. Why is there a discrepancy here? Further, Section 2c also says that a requirement is for the 95% confidence interval of the "difference" in variance unexplained to be within 1% of the mean square wind speed as well. Is this requirement met by the choice of |K| = 500?*

This is a good point. Var(tilde E) = 0.1 was a typo. For increased clarity we replaced this sentence with a reference to the results table. It now reads:

Tables 1 and 2, show that this sample size achieves the goal of 1% relative error with 95% confidence for both Q and ΔQ that we formulated in Section 2 (c). The only exception is the Nearest Neighbors model, for which the confidence interval for the error estimate is 2%. This is still sufficient accuracy for our purpose.

Comment 10 (Figs. 9 and 10) *Consider adding a colorbar to these plots. Further, Fig. 9 says that the distribution is estimated from 1000 steps of the algorithm. Does this mean the full algorithm was run 1000 times (with B = 500), or was it run once using B = 1000?*

We feel that a colorbar is not appropriate in this case, since Figure 5 and 6 have different ranges. With regards to your second point, we changed the figure description to avoid ambiguity. It now reads:

Algorithm 1 was run with $B = 1000$, collecting the Fourier frequencies at each step into a histogram.

Comment 11 (Pg. 16, ln. 39) *"likely improve the accuracy of this approximation" Can you clarify what approximation you are referring to here?*

Changed the sentence to:

If we were to transition from a 2D to a 3D setting by incorporating elevation as a feature in the random Fourier features, divergence penalisation would likely be more useful.

Comment 12 (Pg. 17, ln. 7) *"we derived an optimal density for the Fourier frequencies" As I understand it, you derived a density that minimizes an upper bound to the loss function, but it is not necessarily "optimal". Should this be rephrased?*

We rephrased this sentence as follows:

we derived an upper bound for the mean square error of the random Fourier Frequencies model, found a density that minimised this bound and devised an adaptive Metropolis algorithm for sampling from this density.

Comment 13 (Eq. 7.3) *This loss function doesn't match the loss function used in the main part of the manuscript (Eq. 3.4). Please explain how this proof addresses the specific loss function (with Sobolev norm) in Eq. 3.4.*

We include a sentence before the proof on how to generalise the proof to sums of regularisers:

By replacing $\lambda\bar{C}$ in (7.7) with $\sum_{s=1}^S \lambda_s \bar{C}_s$, Proposition 1 can be generalised to include a linear combination of S regularisation functions $\sum_{s=1}^S \lambda_s \|\mathcal{L}_s \beta\|^2$.

Furthermore, we add a definition (1) of the Sobolev norm that clarifies its connection to the one formulated in Proposition 1:

The expression $\|\beta\|_{S(2,2)}^2$ denotes the second order squared Sobolev-norm of β [1]:

$$\|\beta\|_{S(2,2)}^2 = \|\beta\|^2 + r (\|\partial_x \beta\|^2 + \|\partial_y \beta\|^2) + r^2 (\|\partial_{xx} \beta\|^2 + 2\|\partial_x \partial_y \beta\|^2 + \|\partial_{yy} \beta\|^2) \quad (1)$$

Comment 14 (Pg. 18, ln. 46) *"is exactly the variance of beta" Isn't this actually the variance of the error between beta and f(x)?*

It is both, since beta is unbiased by construction in this part of the proof. To clarify what we mean by the variance of β , we made some slight changes to this part of the proof:

β is then a function of the random variables \mathbf{x} and $\boldsymbol{\omega}$, which we denote $\beta(\mathbf{x};\boldsymbol{\omega})$. With this definition, β is unbiased given \mathbf{x} :

$$E \left[\beta(\mathbf{x};\boldsymbol{\omega}) \mid \mathbf{x} \right] = \sum_{\boldsymbol{\omega} \in Z^2} K \frac{\hat{f}(\boldsymbol{\omega})}{2\pi K \rho(\boldsymbol{\omega})} e^{i\boldsymbol{\omega} \cdot \mathbf{x}} \rho(\boldsymbol{\omega}) = \frac{1}{2\pi} \sum_{\boldsymbol{\omega}' \in Z^2} \hat{f}(\boldsymbol{\omega}') e^{i\boldsymbol{\omega}' \cdot \mathbf{x}} = f(\mathbf{x}), \quad (2)$$

where we used that the components of $\boldsymbol{\omega}$ are iid. Using independence of the residuals ϵ and the data \mathbf{x} , we see that:

$$E [\|\beta(\mathbf{x};\boldsymbol{\omega}) - \mathbf{u}\|^2 \mid \mathbf{x}] = E [\|\beta(\mathbf{x};\boldsymbol{\omega}) - (f(\mathbf{x}) + \epsilon)\|^2 \mid \mathbf{x}] = E [\|\beta(\mathbf{x};\boldsymbol{\omega}) - f(\mathbf{x})\|^2 \mid \mathbf{x}] + \sigma^2, \quad (3)$$

showing that the expected square error of β as a function of \mathbf{x} is the variance of β with respect to the frequencies $\boldsymbol{\omega}$, plus the variance of the noise."

Comment 15 (Pg. 18, ln. 50) : In the left hand side of the equation, should the expectation be conditioned on x , similar to Eq. 7.11?

In this case, no. The conditional expectation is an important step in simplifying the left hand side however, by using the tower property $E[Y] = E[E[Y|X]]$. This step has been left out to shorten the proof.

Comment 16 (Appendix) I understand the point of condensing the proof to keep the appendix concise, but there are several areas that I do not follow. I hope you can provide some clarification, even if it does not make it into the manuscript. First, in the equation in pg. 18, ln. 50, I do not see where the "1/K" in front of $E[|f(x)|^2]$ comes from.

We apologise for the brevity and as you say, the point is to keep the appendix as short as possible. Here a longer version of the derivation, hopefully it will clear up some of your questions:

$$\begin{aligned} E \left[\left\| \sum_{k=1}^K \beta_k e^{i\omega_k \cdot \mathbf{x}} - f(\mathbf{x}) \right\|^2 \right] & (1) = E \left[\left\| \sum_{k=1}^K \left(\beta_k e^{i\omega_k \cdot \mathbf{x}} - \frac{f(\mathbf{x})}{K} \right) \right\|^2 \right] \\ & (2) = \frac{1}{K^2} E \left[\left\| \sum_{k=1}^K (K \beta_k e^{i\omega_k \cdot \mathbf{x}} - f(\mathbf{x})) \right\|^2 \right] \\ & (3) = \frac{1}{K} E \left[\|K \beta_0 e^{i\omega_0 \cdot \mathbf{x}} - f(\mathbf{x})\|^2 \right] \\ & (4) = K E [\|\beta_0\|^2] - \frac{1}{K} E [\|f(\mathbf{x})\|^2] \\ & (5) = \frac{1}{(2\pi)^2 K} E \left[\frac{\|\hat{f}(\omega_0)\|^2}{\rho(\omega_0)^2} \right] - \frac{1}{K} E [\|f(\mathbf{x})\|^2] \\ & (6) \leq \frac{1}{(2\pi)^2 K} \sqrt{ E \left[\frac{\|\hat{f}(\omega_0)\|^4}{\rho(\omega_0)^4} \right] } - \frac{1}{K} E [\|f(\mathbf{x})\|^2]. \end{aligned}$$

In step (1) we move f inside the sum. In step (2), we factor out $1/K^2$. In step (3), we use the independence of the Fourier frequencies to move the sum outside the expectation without getting a "double sum". In step (4), we have used the property that $E[K\beta e^{i\omega \cdot \mathbf{x}}] = f(\mathbf{x})$ to separate the two terms inside the norm, and then factored out the K^2 from the β -term. In step (5), we use the definition of β_0 , to simplify the left term. In step (6) we use Jenessen's inequality to approximate the left term so that we get an expectation that matches with the term from the regulariser later on.

Comment 17 (Eq. 7.12) *Can you explain how you get from the left hand side to the middle of this inequality?*

Thanks to your question, we noticed a slight mistake in the derivation. We amended this section of the proof, which now reads:

Applying \mathcal{L} to $\beta(\mathbf{x})$ is equivalent to multiplying each term $\widehat{\beta}_{kj}e^{i\omega_k x}$, $j = 1, 2$ of the Fourier series with $\ell_j(\omega_k) = \sum_{m=1}^M c_m (i\omega_{k,1})^{\alpha_{m,1}} (i\omega_{k,2})^{\alpha_{m,2}}$, a multivariate polynomial in ω_k of degree d . Define $r(\omega) = |\ell_1(\omega)|^2 + |\ell_2(\omega)|^2$. Note that r has degree $2d$. It follows that

$$E [|\mathcal{L}\beta|^2] = \int_X E [|\mathcal{L}\beta(\mathbf{x}; \boldsymbol{\omega})|^2 | \mathbf{x}] d\mathbf{x} = K(2\pi)^2 E [|\ell_1(\omega_k)\widehat{\beta}_{k1} + \ell_2(\omega_k)\widehat{\beta}_{k2}|^2]. \quad (4)$$

The first equality comes from switching the order of integration, and in the second we use independence of the Fourier frequencies combined with the size $(2\pi)^2$ of the region $X = [0, 2\pi] \times [0, 2\pi]$. Applying the Cauchy-Schwartz inequality twice on the last expression of (4) results in:

$$K(2\pi)^2 E [|\ell_1(\omega_k)\beta_{k1} + \ell_2(\omega_k)\beta_{k2}|^2] \leq K E [r(\omega) |2\pi\beta|^2] \leq \frac{1}{K} \sqrt{E [r(\omega)^2] E \left[\frac{\|\hat{f}(\omega)\|^4}{\rho(\omega)^4} \right]} \quad (5)$$

Here is a line-by-line explanation of equation 7.12.

$$\begin{aligned}
E [||\mathcal{L}\beta||^2] & \quad (1) = E \left[\int_X |\mathcal{L}\beta(x)|^2 dx \right] \\
& \quad (2) = \int_X E [|\mathcal{L}\beta(x)|^2 | x] dx \\
& \quad (3) = \int_X E \left[\left| \sum_{k=1}^K (\ell_1(\omega_k)\beta_{k1} + \ell_2(\omega_k)\beta_{k2}) e^{i\omega_k \cdot x} \right|^2 \right] dx \\
& \quad (4) = \int_X E \left[K |(\ell_1(\omega_0)\beta_{01} + \ell_2(\omega_0)\beta_{02}) e^{i\omega_0 \cdot x}|^2 \right] dx \\
& \quad (5) = \int_X E \left[K |(\ell_1(\omega_0)\beta_{01} + \ell_2(\omega_0)\beta_{02})|^2 \right] dx \\
& \quad (6) = (2\pi)^2 K E \left[|(\ell_1(\omega_0)\beta_{01} + \ell_2(\omega_0)\beta_{02})|^2 \right] \\
& \quad (7) \leq K E [r(\omega_0) ||2\pi\beta_{02}||^2].
\end{aligned}$$

Step (1) is the definition of the norm $||\mathcal{L}\beta||$. In step (2), we switch order of integration. In step (3), we expand the expression $\mathcal{L}\beta$ as a Fourier series. In step (4), we use independence of the Fourier frequencies. In step (5), we use that $|e^{i\omega \cdot x}|^2 = 1$. In (6), we use that the integrand is independent of x . Lastly, we use Cauchy-Schwarz. We have amended the proof accordingly.

Comment 18 (Pg. 19, ln. 19) *Can you define "r_m"? Without this, it is hard to follow the equation around pg. 19, ln. 25, particularly the last equality.*

We define r_m in the first revision of the manuscript, below equation 7.13 as the coefficients of the polynomial $r(\omega)^2$:

The multivariate polynomial $r(\omega)^2 := \sum_{m=1}^{M'} r_m \omega_1^{\alpha_{m,1}} \omega_2^{\alpha_{m,2}}$ is of order $4d$, i.e. $\alpha_{m,1} + \alpha_{m,2} \leq 4d$.

$r(\omega)$ was in turn defined as $|\ell_1(\omega)|^2 + |\ell_2(\omega)|^2$ above equation 7.12. We added some clarification to the steps.

Comment 19 (Eq. 7.17) *I can't find how the derivative of Eq. 7.16 simplifies to this form, since I end up with a summation term with $p(\omega)^3$ in the denominator, in addition to the one with $p(\omega)^4$ that is shown.*

The term with $p(\omega)^3$ is denoted as "c₂" in our proof. Here is a more detailed derivation that shows what happens to this term.

$$H'(\epsilon) = -3 \left(\sum_{\omega \in Z^2} \frac{\|\hat{f}(\omega)\|^4}{(p(\omega) + \epsilon\delta(\omega))^4} \delta(\omega) \right) \left(\sum_{\omega \in Z^2} p(\omega) \right)^3 + 3 \left(\sum_{\omega \in Z^2} \frac{\|\hat{f}(\omega)\|^4}{(p(\omega) + \epsilon\delta(\omega))^3} \right) \left(\sum_{\omega \in Z^2} p(\omega) \right)^2 \sum_{\omega \in Z^2} \delta(\omega)$$

Evaluating at $\epsilon = 0$:

$$H'(0) = -3 \left(\sum_{\omega \in Z^2} \frac{\|\hat{f}(\omega)\|^4}{p(\omega)^4} \delta(\omega) \right) \overbrace{\left(\sum_{\omega \in Z^2} p(\omega) \right)^3}^{:=c_1} + 3 \underbrace{\left(\sum_{\omega \in Z^2} \frac{\|\hat{f}(\omega)\|^4}{p(\omega)^3} \right)}_{:=c_2} \left(\sum_{\omega \in Z^2} p(\omega) \right)^2 \sum_{\omega \in Z^2} \delta(\omega)$$

We define the constants c_1 and c_2 as above. Then, setting $H'(0) = 0$ we get

$$\sum_{\omega \in Z^2} \left(\frac{\|\hat{f}(\omega)\|^4}{p(\omega)^4} - \frac{c_2}{c_1} \right) \delta(\omega) = 0$$

And since $\delta(\omega)$ is arbitrary, and p sums to 1, we must have

$$\frac{\|\hat{f}(\omega)\|^4}{p(\omega)^4} - \frac{c_2}{c_1} \iff p(\omega) \propto \|\hat{f}(\omega)\|. \iff p(\omega) = \frac{\|\hat{f}(\omega)\|}{\sum_{\omega \in Z^2} \|\hat{f}(\omega)\|}.$$

Minor/formatting

Comment 20 (Pg. 3, ln. 23) "pressure are used pose" -j "used to pose"

Changed to

assumptions about wind field, temperature and pressure are expressed as a set of differential equations that describe how the system evolves over time.

Comment 21 (Fig. 1 caption) *The degree symbol does not show up properly. Similar formatting issues occur throughout the manuscript (e.g., references 8, 19, 27)*

This is due to "special characters" like °,ö,ä,å not showing up properly when compiling the document. We suspect this has to do with the way that the document is compiled in ScholarOne, as it works fine in overleaf. We believe this issue is resolved now.

Comment 22 (Pg. 5, ln. 6) *Here the phrase "hyper parameter" is used, but sometimes it is spelled as one word "hyperparameter". Please be consistent.*

Changed to hyper parameter everywhere.

Comment 23 (Eq. 3.9) *I believe "x" on the left hand side should be bold.*

Changed left hand side "x" to bold

Comment 24 (Pg. 7, ln. 51) *"the random Fourier..." capitalize "the"*

Capitalised "the".

Comment 25 (Pg. 8, ln. 44) *"north-west length" -j "east-west" or "north-south"?*

changed to "north-south".

Comment 26 (Algorithm 1) *The symbol "r", the rounded elements of sigma r_N , is already used to describe a regularization parameter. Can a different symbol be used here?*

Changed to δ_N and δ .

Comment 27 (Eq. 3.15) *The equation uses $x - x$ prime, but the text above uses x prime - x . Just a suggestion to make them consistent.*

Switched the text above to $x - x'$.

Comment 28 (Pg. 11, ln. 58) *"two standard deviations of correspond": two standard deviations of what?*

Changed to:

means that two standard deviations of \mathcal{E} and \mathcal{Q} correspond to 95 % confidence intervals for \mathcal{E} and \mathcal{Q} , respectively.

Comment 29 (Fig. 5 caption) *Capitalize "sweden". Further, the fractions of variance unexplained listed are 0.57, 0.55, 0.7. These don't match the values in Table 1. Are they just the metrics for this particular time step?*

Capitalised Sweden. They are the fraction of variance unexplained for this particular time step. We clarified this:

From left to right, the fraction of variance unexplained for the above methods at this specific time is 0.57, 0.55 and 0.70.

Comment 30 (Pg. 18, ln. 6) *"a family a family" typo*

Fixed.

Comment 31 *The multi line equations in the appendix] are missing equation numbers (pg. 18, ln. 50 and pg. 19, ln. 25).*

Added equation numbers.

Comment 32 (Pg. 20, ln. 24) *Commas are missing between the author names.*

Additional changes

Apart from the above comments we made the following minor changes to the document:

1. we replaced the reference [2] with a published version, [3]
2. We corrected an error in the Funding statement. The funding statement now reads as follows.

Jonas Kiessling and Raúl Tempone were partially supported by the KAUST Office of Sponsored Research (OSR) under Award numbers OSR-2019-CRG8-4033 and OSR-2019-CRG8-4033.2, and the Alexander von Humboldt Foundation, through the Alexander von Humboldt Professorship award. Emanuel Ström was supported by the KAUST Visiting Student Research Program (VSRP).

References

- [1] Philippe Blanchard and Erwin Bruning. *Mathematical Methods in Physics*, volume 69 of *Progress in Mathematical Physics*. Birkhäuser, 2nd edition, 2015.
- [2] Luis Espath, Dmitry Kabanov, Jonas Kiessling, and Raúl Tempone. Statistical learning for fluid flows: Sparse fourier divergence-free approximations, 2021.
- [3] Luis Espath, Dmitry Kabanov, Jonas Kiessling, and Raúl Tempone. Statistical learning for fluid flows: Sparse fourier divergence-free approximations. *Physics of Fluids*, 33(9):097108, 2021.